# Discovery of Novel Coumarin Derivatives as Potential Dual Inhibitors against α-Glucosidase and α-Amylase for the Management of Post-Prandial Hyperglycemia via Molecular Modelling Approaches

**DOI:** 10.3390/molecules27123888

**Published:** 2022-06-17

**Authors:** Shashank M. Patil, Reshma Mary Martiz, A. M. Satish, Abdullah M. Shbeer, Mohammed Ageel, Mohammed Al-Ghorbani, Lakshmi Ranganatha, Saravanan Parameswaran, Ramith Ramu

**Affiliations:** 1Department of Biotechnology and Bioinformatics, School of Life Sciences, JSS Academy of Higher Education and Research, Mysuru 570015, India; shashankmpatil@jssuni.edu.in (S.M.P.); reshmamarymartiz@jssuni.edu.in (R.M.M.); saravananp@jssuni.edu.in (S.P.); 2Department of Pharmacology, JSS Medical College, JSS Academy of Higher Education and Research, Mysuru 570015, India; amsatish@jssuni.edu.in; 3Department of Surgery, Faculty of Medicine, Jazan University, Jazan 45142, Saudi Arabia; ashbeer@jazanu.edu.sa (A.M.S.); ageelahmed@jazanu.edu.sa (M.A.); 4Department of Chemistry, College of Science and Arts, Taibah University, Madina 41477, Saudi Arabia; malghorbani@yahoo.com; 5Department of Chemistry, College of Education, Thamar University, Thamar 425897, Yemen; 6Department of Chemistry, The National Institute of Engineering, Mysuru 570008, India; lranganath.v@gmail.com

**Keywords:** coumarin, α-glucosidase, α-amylase, postprandial hyperglycemia, diabetes mellitus, molecular modeling

## Abstract

Coumarin derivatives are proven for their therapeutic uses in several human diseases and disorders such as inflammation, neurodegenerative disorders, cancer, fertility, and microbial infections. Coumarin derivatives and coumarin-based scaffolds gained renewed attention for treating diabetes mellitus. The current decade witnessed the inhibiting potential of coumarin derivatives and coumarin-based scaffolds against α-glucosidase and α-amylase for the management of postprandial hyperglycemia. Hyperglycemia is a condition where an excessive amount of glucose circulates in the bloodstream. It occurs when the body lacks enough insulin or is unable to correctly utilize it. With open-source and free in silico tools, we have investigated novel 80 coumarin derivatives for their inhibitory potential against α-glucosidase and α-amylase and identified a coumarin derivative, CD-59, as a potential dual inhibitor. The ligand-based 3D pharmacophore detection and search is utilized to discover diverse coumarin-like compounds and new chemical scaffolds for the dual inhibition of α-glucosidase and α-amylase. In this regard, four novel coumarin-like compounds from the ZINC database have been discovered as the potential dual inhibitors of α-glucosidase and α-amylase (ZINC02789441 and ZINC40949448 with scaffold thiophenyl chromene carboxamide, ZINC13496808 with triazino indol thio phenylacetamide, and ZINC09781623 with chromenyl thiazole). To summarize, we propose that a coumarin derivative, CD-59, and ZINC02789441 from the ZINC database will serve as potential lead molecules with dual inhibition activity against α-glucosidase and α-amylase, thereby discovering new drugs for the effective management of postprandial hyperglycemia. From the reported scaffold, the synthesis of several novel compounds can also be performed, which can be used for drug discovery.

## 1. Introduction

Type 2 diabetes mellitus (T2DM) and its subsequent complications are realized as a severe endocrine health concern, where the altered body metabolism occurs due to the imbalance in insulin levels [1]. Hyperglycemia, or high blood sugar levels, is caused by the deficient amount of insulin produced by the deformed beta-cells of the pancreas, which increases the risk of T2DM [2,3]. In this context, carbohydrate digestive enzymes (α-glucosidase and α-amylase) play a crucial role in the elevation of hyperglycemia. They accelerate the release of monosaccharides via hydrolyzing the α-(1→4) bonds between oligosaccharides. This causes the elevation in blood sugar levels, which in turn increases the risk of T2DM [4]. α-glucosidase resides in the epithelial cells of the small intestine [5], whereas α-amylase is predominantly present in the pancreas, but later released into the small intestine [6,7]. Both the enzymes are the chief causative factors of postprandial hyperglycemia, as the digestion of carbohydrates occurs in the small intestine [7]. As a result, α-glucosidase and α-amylase inhibition has been regarded as one of the most effective strategies to treat T2DM [1,4].

A variety of chemotherapeutics are utilized to inhibit the enzymatic action of α-glucosidase and α-amylase. These oral hypoglycemic medications are mostly linked to gastro-intestinal abnormalities, which affect more than 50% of patients. Colonic gas is produced as unabsorbed carbohydrates ferment, resulting in abdominal bloating, increased flatulence, cramping, and diarrhea. These chemotherapeutics are not recommended for people with acute or chronic gastrointestinal problems. In addition, they are also reported with renal impairment and hepatic dysfunction [8]. Therefore, the understanding of the adverse effects of these agents paves the way for effective chemotherapeutics.

The coumarins are heterocyclic compounds belonging to the class of benzopyrone enriched in various plants such as tonka beans (*Dipteryx odorata*), Chinese angelica (*Angelica dahurica*), Pink Lime-Berry (*Clausena excavata*), Silk cotton tree (*Bombax ceiba*), Chamomile (*Matricaria chamomilla*) and others [9]. Generally, coumarins and their derivatives are plant-derived natural products, microorganism-derived metabolites, and synthetic molecules, known for their pharmacological properties such as anticoagulant, antifungal, anti-inflammatory, antiviral, antibacterial, antihypertensive, anticancer, anticonvulsant, antitubercular, antiadipogenic, antioxidant, antihyperglycemic, and neuroprotective properties [10]. Most of these effects can be attributed to their free radical scavenging effects. Coumarins such as umbelliferone, esculetin, and quercetin show antioxidant properties and protect the cellular DNA from oxidative damage. The dicumarol shows anticoagulant properties by inhibiting the action of vitamin K, whereas angelmarin has been reported to be cytotoxic in pancreatic cancer [11].

Further, the dietary exposure to benzopyrones is significant as these compounds are found in fruits, seeds, nuts, vegetables, coffee, tea, and wine. In view of the established low toxicity, relative cost-effectiveness, presence in the diet, and occurrence in various herbal remedies of coumarins, it seems prudent to evaluate their pharmacological properties and therapeutic applications [12,13,14]. To date, many of the coumarins have been isolated from higher plants; some of them have been discovered in microorganisms. The important coumarin members belonging to microbial sources are novobiocin, coumermycin, and chartreusin [15]. Among synthetic coumarins, hydroxyl aromatic substituted derivatives such as 5-hydroxycoumarin or vicinal dihydroxy coumarins have also been found to be potent anti-inflammatory agents [16].

Thus, it has been proven that natural/synthetic coumarins have a huge potential to serve as drugs for various diseases or disorders due to their extensive pharmacological properties, which fascinates many medicinal chemists for further backbone derivatization and screening them as various novel therapeutic agents. Considerable research and efforts are needed to bring coumarins to the stage of clinical trials for further approval and tap their potential for translational research. We have recently synthesized novel coumarin derivatives and are keen to evaluate their antidiabetic potential. In addition, the structural information of the derivatives has been experimentally determined. Meanwhile, we want to investigate their antidiabetic potential utilizing bioinformatics approaches so that we can design our biological studies in a rational manner. Computational methods such molecular docking, molecular dynamics simulation, and binding free energy calculations have been proven to provide reliable predictions in the arena of drug discovery. They save a tremendous amount of time and wealth spent on in vitro and in vivo studies. Therefore, the current study aims to discover our novel coumarin derivatives as potential inhibitors against both α-glucosidase and α-amylase through computational approaches for the management of postprandial hyperglycemia. In the near future, we aim to evaluate the biological activity of these compounds through in vitro and in vivo approaches, based on the outcomes of the present study.

## 2. Materials and Methods

### 2.1. Preparation of Protein Target and Ligands

Since the aim of this work is to develop dual inhibitors for human α-glucosidase and human α-amylase enzymes, we searched for the same using RCSB PDB database (https://www.rcsb.org/, accessed on 28 April 2022). On the other hand, human α-glucosidase is not yet characterized. Therefore, we had to retrieve the yeast α-glucosidase (PDB ID: 3A4A; chain A) from RCSB PDB database. As the authors have already proven the inhibition of yeast α-glucosidase in vitro using umbelliferone (IC_50_: 7.08 ± 0.17 μg/mL) and lupeol (IC_50_: 7.18 ± 0.14 μg/mL), selection of yeast α-glucosidase for this study was found to be in accordance with the previous studies [17,18]. Yet, the human and human α-amylase (PDB ID: 2QV4; chain A) was retrieved from RCSB PDB database.

Preparation of protein molecules was carried out with standard protocol using AutoDockTools1.5.6 as described in our previous study by Patil et al. (2021) [8]. Further, the inhibitor binding site of substate/co-crystal ligand/inhibitor was elucidated from the literature for α-glucosidase [19] and α-amylase [20]. The grid box of size 40 Å × 40 Å × 40 Å was built that covers the inhibitor binding pocket for α-glucosidase. Similarly, the grid box of 30 Å × 30 Å × 30 Å was built for α-amylase. The 2D chemical structures of coumarin derivatives were drawn and their 3D structure was optimized using ACD ChemSketch. Further, the ligand preparation for docking was carried out with AutoDockTools1.5.6 as described in our previous study by Patil et al., (2021) [8].

### 2.2. Virtual Molecular Docking of Coumarin Derivatives and Its Validation with Molecular Dynamics Simulations

Virtual screening of the ligands was performed using AutoDock Vina 1.1.2. The 3D conformations of the coumarin derivatives obtained from docking were screened based on their binding affinity, total number of intermolecular bonds, and hydrogen bonds with their respective drug targets, α-glucosidase and α-amylase [21]. Based on these parameters, a potential dual inhibitor (single coumarin derivative, herein referred to as HIT1) with better free energy of binding and highest number of interactions for both the drug targets since the study aims to discover a new common inhibitor for both the enzymes. The reliability of the docking studies was investigated with molecular dynamics (MD) simulation with explicit water model to predict the preferential binding of HIT1 with their respective drug targets in the presence of water. Acarbose was considered a positive control owing to its inhibiting potential of both the drug targets. The drug is known to reduce hyperglycemic levels and is popular among diabetic individuals. It was also used as a positive control in our previous studies [8,22,23]. The biomolecular software package, GROMACS 18.1 was employed to perform molecular dynamics simulations. The ligand structures were approximated by the CHARMM36 force field and ligand topology was generated using SwissParam server. On the other hand, protein structure was also added with the CHARMM36 forcefield using the pdb2gmx module. This was followed by energy minimization in vacuum of 5000 steps with steepest descent method. Each protein complex was placed in a box at 10 Å distance to the edges. TIP3P water model was used to incorporate the solvent, to which appropriate number of Na^+^ and Cl^−^ counter ions were added to maintain the essential 0.15 M salt concentration. In total, 2 apo-structures and 2 protein–ligand complexes were prepared for simulation. Energy minimization of the apo-structures and complexes was performed using the steepest descent followed by conjugate gradient methods. A brief equilibration of NPT ensemble (1000 ps) and NVT ensemble (1000 ps) was carried out to optimize the molecular systems for the production runs. All the production run simulations were run for 100 ns at 310 K temperature and 1 bar pressure [23]. Using XMGRACE, trajectory plots from the simulation including root mean square deviation (RMSD), root mean square fluctuation (RMSF), radius of gyration (Rg), solvent accessible surface area (SASA) and intermolecular hydrogen bonds between HIT1 and respective drug targets were obtained [24].

### 2.3. Pharmacophore Studies

The PharmaGist webserver was used to generate a ligand-based 3D pharmacophore based on the refined binding pose of HIT1 from MD simulations to discover pharmacophoric features by coumarin-like derivates and new molecular scaffolds for the dual inhibition of the drug targets of interest [25]. Among the several outcomes obtained, appropriate pharmacophoric features were chosen from the webserver independently that might lead to the inhibition of the respective drug targets of interest, α-glucosidase and α-amylase. We used the ZINCPharmer web application to screen the molecules from the ZINC database utilizing the refined pharmacophoric features of α-glucosidase and α-amylase independently, followed by docking studies of the screened molecules against the respective drug targets of interest [25].

### 2.4. Molecular Docking and Dynamics Simulation of Selected Compounds

The compounds obtained from pharmacophore studies were investigated for the binding with both the drug targets of interest followed by the validation with the molecular dynamics simulations as described earlier to discover a potential dual inhibitor against both α-glucosidase and α-amylase [8,21].

### 2.5. Binding Free Energy Calculations

From the outcomes of molecular dynamic simulations run for selected coumarin derivatives, binding free energy calculations were calculated using molecular mechanics/Poisson–Boltzmann surface area (MM-PBSA) technique [26]. The binding free energy for each ligand–protein complex was calculated using the g_mmpbsa program with MmPbSaStat.py script, which utilizes the GROMACS 2018.1 trajectories from the last 50 ns as input. The binding free energy was calculated using three components: molecular mechanical energy, polar and apolar solvation energies, and molecular mechanical energy. The calculation was performed with the trajectories to compute ΔG with 1000 configurations of the protein–ligand complexes. The molecular mechanical energy and polar and apolar solvation energies are used to evaluate the ΔG. The free binding energy is calculated using Equations (1) and (2).
ΔG_Binding_ = G_Complex_ − (G_Protein_ + G_Ligand_)(1)
ΔG = Δ_EMM_ + ΔG_Solvation_ − TΔS = ΔE_(Bonded + non-bonded)_ + ΔG_(Polar + non-polar)_ − TΔS(2)

G_Binding_: binding free energy, G_Complex_: total free energy of the protein–ligand complex, G_Protein_ and G_Ligand_: total free energies of the isolated protein and ligand in solvent, respectively, ΔG: standard free energy, Δ_EMM_: average molecular mechanics potential energy in vacuum, G_Solvation_: solvation energy, ΔE: total energy of bonded as well as non-bonded interactions, ΔS: change in entropy of the system upon ligand binding, and T: temperature in Kelvin [27,28].

## 3. Results and Discussion

### 3.1. Virtual Screening of Ligands by Docking and Dynamics Simulation

Virtual screening of the coumarin derivatives against α-glucosidase and α-amylase resulted in all the compounds being bound to the inhibitor binding site of both the target enzymes. However, coumarin derivative-59 (CD-59) was predicted with the highest binding affinity (better predicted free energy of binding), the total number of intermolecular interactions including hydrogen bonds in the case of both α-glucosidase and α-amylase. Since the current study was designed to find a dual inhibitor, CD-59 was selected for further in silico analyses as the coumarin derivative has the above-mentioned criteria compared to acarbose and other coumarin derivatives for both the drug targets. Table 1 provides the structural details about the coumarin derivatives used in this study, whereas the details of virtual screening of the coumarin derivatives have been given in Table 2 (selected potential dual inhibitors) and Appendix A (all the coumarin derivatives docked).

In the case of α-glucosidase, the coumarin derivative CD-59 was able to bind to the inhibitor binding site of the protein molecule, where the co-crystallized inhibitor compound (α-D-glucopyranose) was bound. The binding affinity (predicted free energy of binding) of CD-59 was found to be −11.6 kcal/mol. It formed a total of 14 non-bonded intermolecular interactions including four hydrogen bonds via Gly161, His423, Arg315, and Gly160. Hydrophobic pi–pi stacked bonds including Tyr158 and Phe314 were formed. Alkyl bond with Ala418 and pi–alkyl bond with Phe178 were formed. In addition, a pi–anion bond with Asp233 and a pi–cation bond with Lys156 were also formed. With the above-mentioned interactions, CD-59 was primarily bound with the domain A of the enzyme, within the inhibitor binding site. These binding interactions resembled the interactions reported in previous reports by Peytam et al. (2021) [29] where 3-amino-2,4-diarylbenzo [4,5] imidazo [1,2-a] pyrimidines were evaluated for their α-glucosidase (PDB ID: 3A4A) inhibitory activity (used AutoDock Tools 1.5.6 for docking). Additionally, the penetration of the CD-59 into the inhibitor binding site leads to the formation of more binding interactions in comparison with Shukla et al. (2021) [30], who evaluated glycosyl-1, 2, 3-1H-triazolyl methyl benzamide derivatives as the α-glucosidase (PDB ID: 3A4A) inhibitors (used Schrodinger Glide module for docking). Compared to both studies, the coumarin derivative CD-59 had a higher binding affinity and significant intermolecular interactions. With the key residues of α-glucosidase reported in the above-mentioned studies, the coumarin derivative CD-59 can act as an effective inhibitor of α-glucosidase. However, the reference compound acarbose was predicted with only five interactions, with all of them being hydrogen bonds. Acarbose was also bound with the domain A of the enzyme. It interacted with Glu411, Asp307, Pro312, Thr310, and His280, where Thr310 and His280 were found to be donor–donor unfavorable interactions. The binding affinity of acarbose was −7.9 kcal/mol. From the perspective of the software used, our study resulted in the same binding interaction pattern as Shukla et al. (2021) [30], even though we have used open-source software such as AutoDock Vina 1.1.2., and the latter used the commercial software (Schrodinger Glide module). Therefore, the open-source software AutoDock Vina 1.1.2. is also reliable and equivalent in terms of its efficiency to the commercial software Schrodinger Glide module. The visualization of the binding interaction of CD-59 and acarbose with α-glucosidase is given in Figure 1 highlights that the coumarin derivative, CD-59, has more binding affinity and inhibiting potential than acarbose via significant intermolecular interactions.

In the case of α-amylase (PDB ID 2QV4) complexed with acarbose, CD-59 was predicted to bind within the inhibitor binding site, where the co-crystal ligand acarbose was bound. It was bound with the key residues with Asp300 (hydrogen bond) and Glu233 (pi–anion), which leads to better inhibition. Apart from these interactions, CD-59 interacted with Gln63 and Al198 with hydrogen bonds, Trp59 and Tyr151 with pi–pi bonds, Leu162 with pi–alkyl bonds, and Ile235 with pi–sigma bonds. In total, it formed 16 intermolecular interactions including three hydrogen bonds with a binding affinity of −11.3 kcal/mol. The binding of the CD-59 into the key residues was in accordance with the previous study by Swilam et al. (2022) [31], who reported polyphenols from *Ammannia baccifera* as the potential α-amylase inhibitors (who used Molecular Operating Environment/MOE 2019.01). Binding interactions of CD-59 were found to be better in comparison with Mor and Sindhu (2019) [32] (who used AutoDock Vina 1.1.2), and Hajlaoui et al., (2021) [33] (who used AutoDock Vina 1.1.2) with respect to the interaction with key residues of the inhibitor binding site. From the above-mentioned outcomes, it becomes evident that CD-59 could act as a potential inhibitor of the α-amylase enzyme. Meanwhile, acarbose formed only two hydrogen bonds with Gln63 and Thr163, with the bond connecting Gln63 being unfavorable (donor–donor) with a binding affinity of −7.7 kcal/mol. From the perspective of the software used, outcomes from our study are in accordance with the reported studies, where AutoDock Vina 1.1.2 was used. The visualization of the binding interaction of CD-59 and acarbose with α-amylase is given in Figure 2.

A molecular dynamics simulation was used to validate the docking investigation and determine the degree of stability of the docked complex along with the target protein. Therefore, it becomes essential to perform molecular dynamics simulation after the docking simulation (Santos et al., 2019) [34]. MD trajectories are represented in Figure 3 for CD-59 and acarbose bound with α-glucosidase along with the wild-type (apo-structure) of α-glucosidase. The root mean square deviation (RMSD) graph represents the protein–CD-59 complex’s stability throughout the course of a 100 ns simulation. By examining the plot, it can be said that both CD-59 and acarbose stayed inside the inhibitor binding site throughout the simulation period, and never came out of the site. However, the protein–acarbose complex was never found to obtain stabilization until 20 ns, whereas the protein–CD-59 and apoprotein were stabilized after 10 ns. In the RMSF analysis, both the CD-59 complex and the apoprotein were on par, with an almost similar fluctuation pattern. In the protein–acarbose complex, a higher number of fluctuations was observed in the loop regions (200–250 residues). Fluctuations were also found between 100–150 regions. Higher fluctuations in RMSF plots indicate the instability of acarbose inside the inhibitor binding site. Further, Rg and SASA plots were analyzed to show the structural compactness of the structure formed. The Rg plot analysis shows that the protein–CD-59 complex was compact throughout the simulation, which resulted in the decrease in the SASA value, as the occupancy of the ligand increased. Finally, the ligand hydrogen bond was assessed to determine the structural pre-agreement, and it can be seen that the complex may have undergone structural alterations. However, in ligand hydrogen bond formation, acarbose formed more hydrogen bonds (nine) in comparison with CD-59 (seven). The MD simulation outcomes of CD-59 were in accordance with the previous study, where stilbene derivatives were docked and simulated with the α-glucosidase (PDB ID: 3A4A) (Lee et al., 2014) [35]. Although Lee et al., (2014) [35] and Liu et al., (2020) [36] have performed the MD simulations for the same protein target, both the studies represent the stability only through the RMSD plots, whereas our study reports the stability through various plots obtained from GROMACS 18.1. The outcomes from MD simulation of CD-59 and acarbose complexed with the protein show that both CD-59 and acarbose penetrated at the inhibitor binding site and perform stable interactions that might contribute to their inhibitory activity. However, CD-59 was found to be comparatively better than acarbose in all the parameters considered for simulation studies. Table 3 represents the MD trajectory values of CD-59 and acarbose complexed with α-glucosidase.

In the case of α-amylase, the RMSD plots depict that the CD-59 and acarbose did not get separated from the inhibitor binding site throughout the simulation. However, protein–acarbose was not stabilized until 30 ns, whereas protein–CD-59 and backbone atoms were stabilized after 10 ns. In the RMSF analysis, both the CD-59 complex and the apoprotein were in accordance, with an almost similar fluctuation pattern. In the protein–acarbose complex, a higher number of fluctuations was observed in the loop regions (250–300 residues) and between 100–150 residues, which depict the instability of acarbose inside the inhibitor binding site. Further, Rg and SASA plots were analyzed to show the structural compactness of the structure formed. The Rg plot analysis shows that the protein–CD-59 complex was compact throughout the simulation period and the SASA value was found to be in a similar pattern. However, while investigating the intermolecular interactions, acarbose formed more hydrogen bonds (seven) in comparison with CD-59 (six) during the simulation. The MD simulation run for α-amylase complexed with CD-59 was in accordance with the previous results obtained in [37,38,39]. The stability of CD-59 inside the inhibitor binding site of the protein was hence proven in comparison with the published literature. With the overall stability observed in MD trajectory analysis, CD-59 could act as a potential α-amylase inhibitor. The visualization of dynamics simulation trajectories for α-amylase has been depicted in Figure 4. The MD trajectory values obtained for CD-59 and acarbose complexed with α-amylase are given in Table 4.

### 3.2. Pharmacophore Studies

The ligand-based pharmacophoric features for the drug targets of interest, α-glucosidase, and α-amylase were detected independently based on their respective chosen configurations from the MD simulations in complex with the CD-59 (HIT1) with PharmaGist webserver. Though thirteen pharmacophoric features were detected for CD-59 for both α-glucosidase and α-amylase, their pharmacophoric features were different due to their differential binding obtained from their respective MD simulations. It is noteworthy to mention that CD-59 binds to α-amylase in an extended form while it binds to α-glucosidase in a non-extended and bent form. The key pharmacophoric features were selected to perform virtual screening with ZINCPharmer against the ZINC database since the consideration of all possible thirteen features did not screen any molecules from the ZINC database. The pharmacophoric model chosen to screen molecules against α-glucosidase was presented in Figure 5, which comprises four aromatic rings and two H-bond donors that are contributed by four six-membered rings and two double-bonded oxygen atoms. Meanwhile, the authors have chosen a slightly different pharmacophore model for α-amylase by replacing the first aromatic ring feature from the double six-membered ring with a five-membered ring to increase the diversity of the screened molecules. The choice of pharmacophoric features is to ensure that the pharmacophore screening yield one H-bond and most of the hydrophobic interactions that could anchor the screened molecules in the respective ligand-binding pockets of α-glucosidase and α-amylase. Results obtained in pharmacophore modeling were in accordance with the previous study by Chenafa et al., (2021) [40], which evaluated selected phytochemicals as the dual inhibitors of α-glucosidase and α-amylase.

Herein, we report that CD-59 is a potent molecule for the inhibition of α-glucosidase and α-amylase by the Insilco method. In the CD-59 molecule, the stereochemical arrangement of the substituents (bromo and methyl groups) gives more stability to the molecule. Here, the bulkier bromo group is present at the second/ortho position of the phenyl ring, which is diagonal to the methyl group present at the para position of the benzoyl ring. Due to the high molecular weight and electronegative effect of the bromo group, which in turn forms highly stable intermolecular hydrogen bonding with α-glucosidase and α-amylase. This strong interaction between the ligand and protein leads to more binding energy with good interaction. Further, the methyl group is present at the para position of the phenyl ring of benzophenone, which is planar and stereochemical stable; this factor is highly responsible for the potentiality of the molecule for the inhibition of α-glucosidase and α-amylase enzymes. During the molecular docking of CD-59, the methyl group formed a hydrophobic pi–alkyl interaction with Phe178, whereas the bromo group formed an electrostatic pi–cation interaction with Lys156 of α-glucosidase (Figure 1). In the case of α-amylase, the methyl group formed a hydrophobic pi–pi stacked interaction with Trp59 (Figure 2). These interactions ensure that both the bromo and methyl groups play a crucial role in the binding interaction of CD-59.

### 3.3. Molecular Docking and Dynamics Simulation of Selected Coumarin Derivatives

In the case of α-glucosidase, molecular docking simulation was carried out for the compounds obtained from pharmacophore results obtained for α-glucosidase (107 compounds) and α-amylase (55 compounds). The results of the virtual screening of pharmacophore compounds for α-glucosidase and α-amylase have been given in Table 5 and Table 6, respectively. Out of these results, four of the pharmacophore compounds were selected as potential dual inhibitors both α-glucosidase and α-amylase (Table 7). In the case of Table 5 (α-glucosidase inhibitors), compounds with a binding affinity of more than −11.0 kcal/mol were retained. The details of the other compounds (with a binding affinity less than −11.0 kcal/mol) have been given in Appendix A. The same criteria were applied to Table 6 (α-amylase inhibitors), where compounds with a binding affinity of more than −9.5 kcal/mol were retained. The details of the other compounds (with a binding affinity less than −9.5 kcal/mol) have been given in Appendix A. They include ZINC02789441, ZINC40949448, ZINC13496808, and ZINC09781623. These were considered the potential dual inhibitors of α-glucosidase and α-amylase.

Both ZINC02789441 and ZINC40949448 were predicted with the same scaffold (thiophenyl chromene carboxamide), whereas ZINC13496808 was predicted with triazino indol thio phenylacetamide, and ZINC09781623 was predicted with chromenyl thiazole as their scaffolds (Figure 6). Out of these compounds, ZINC02789441 was predicted with the highest binding affinity and most intermolecular and hydrogen bond interactions [8,41].

ZINC02789441 formed a total of 15 non-bonding intermolecular interactions including four hydrogen bonds with a binding affinity of −11.4 kcal/mol. Similar to the compound CD-59, ZINC02789441 was bound within the inhibitor binding site (domain A) of the enzyme. It formed hydrogen bonds with Arg315, Thr310, and His280. It also formed a pi–pi hydrophobic bond with Tyr158, followed by a pi–anion bond with Asp307. It also formed a pi–alkyl bond with Pro312, Val308, and Ala329, followed by an alkyl bond with Lys156. Since the binding interaction of ZINC02789441 was on par with that of CD-59, it becomes evident that ZINC02789441 could act as a potential inhibitor of α-glucosidase. Even the previously reported studies also depict the same binding pattern [42,43]. Acarbose was predicted with only five hydrogen bonds with the domain A of the enzyme. It interacted with Glu411, Asp307, Pro312, Thr310, and His280, with Thr310, and His280 having donor–donor unfavorable interactions. The binding affinity of acarbose was −7.9 kcal/mol. The binding interaction pattern of acarbose was found to be the same, as reported during the virtual first screening of the coumarin derivatives against α-glucosidase (Table 2). The visualization of the binding interaction of ZINC02789441 and acarbose with α-glucosidase is given in Figure 7.

In the case of α-amylase, out of all the compounds screened, ZINC02789441 was found to have the highest binding affinity, non-bonding intermolecular interactions, and hydrogen bonds. Even though the other compounds showed higher binding affinity, we had to choose ZINC02789441 because the aim of this study is to choose a dual inhibitor of both enzymes. With a binding affinity of −9.5 kcal/mol, 16 non-bonding intermolecular interactions, and 1 hydrogen bond, ZINC02789441 bound to the key residue Glu233 with 2 pi–anion bonds. Even with a binding affinity of −9.5 kcal/mol, we had to select ZINC02789441 as the most potent inhibitor, as the aim of the study was to discover dual inhibitors of the target enzymes. The other compounds were found to deviate toward single enzyme inhibition. The binding pattern of ZINC02789441 was found to be similar to that of the CD-59 and was in accordance with the previously reported studies [44,45]. Thus, binding to the reported key residues could represent ZINC02789441 as a potential α-amylase. Since the binding efficiency of the ZINC02789441 has already been proven in the case of α-glucosidase, it could act as a dual inhibitor of both the enzymes. Yet, acarbose was bound with two hydrogen bonds (Gln63 and Thr163). The bond connecting Gln63 was found to be unfavorable (donor–donor). A binding affinity of −7.7 kcal/mol was predicted for acarbose. The binding interaction pattern of acarbose was found to be the same, as reported during the virtual first screening of the coumarin derivatives against α-amylase (Table 2). The visualization of the binding interaction of ZINC02789441and acarbose with α-amylase is given in Figure 8.

Molecular dynamics simulation of ZINC02789441 with α-glucosidase reveals that the RMSD plot of apoprotein and protein ZINC02789441 were in accordance with each other. It shows that ZINC02789441 was present inside the inhibitor binding site of α-glucosidase throughout the simulation. The RMSF plots show that the protein–acarbose complex was predicted with higher fluctuations in the loop regions as well as the *N*-terminal region, indicating the instability of acarbose inside the active site. However, Rg plots show that both the protein–ligand complexes are compact. A similar pattern of results was obtained for SASA plots. In the case of ligand hydrogen bonds, acarbose was predicted with lesser hydrogen bonds (seven) in comparison with ZINC02789441 (nine). ZINC02789441 was predicted with a higher number of hydrogen bonds compared to CD-59. However, ZINC02789441 formed more hydrogen bonds compared to CD-59. The MD trajectories obtained for ZINC02789441 were found to be similar to those of CD-59, as well as previously reported studies [30,35]. The visualization of MD trajectories for both ZINC02789441 and acarbose complexed with α-glucosidase are given in Figure 9. The values of MD trajectories for ZINC02789441 and acarbose complexed with α-glucosidase are detailed in Table 8.

In addition to α-glucosidase, ZINC02789441 was simulated with α-amylase. The RMSD plots show that ZINC02789441 was present in the protein inhibitor binding site throughout the simulation period. In the case of RMSF plots, the protein–acarbose complex showed higher fluctuations throughout the simulation, including both terminal regions and loop regions, indicating the instability of acarbose. The same pattern of results was observed in the case of Rg and SASA plots. In addition, ZINC02789441 formed more hydrogen bonds (seven) than acarbose (six) during MD simulation. ZINC02789441 even formed more hydrogen bonds than CD-59. The MD trajectories obtained for ZINC02789441 were in accordance with those of CD-59 and previously reported studies [46]. The visualization of MD trajectories for both ZINC02789441 and acarbose complexed with α-amylase is depicted in Figure 10. The values of MD trajectories for ZINC02789441 and acarbose complexed with α-amylase are given in Table 9.

### 3.4. Binding Free Energy Calculations

The binding free energy analysis of ZINC02789441 reveals that van der Waal’s energy and binding energies have played a significant role in the formation of protein–ligand complexes during MD simulation. All the binding free energy calculations for ZINC02789441 were energetically feasible. In comparison with ZINC02789441 bound complexes, acarbose bound complexes showed lesser binding free energies, indicating the comparatively weaker protein–ligand interactions and binding affinity. Results from binding affinity support the outcomes from molecular docking and dynamics simulation. Additionally, these outcomes were on par with the previous studies, which have performed binding free energy calculations for α-glucosidase and α-amylase [5,35]. Table 10 summarizes the results of binding free energy calculations obtained using the MMPBSA technique.

## 4. Conclusions

The search for the potential inhibitors of α-glucosidase and α-amylase inhibitors has been the center of focus since both the carbohydrate digestive enzymes are associated with post-prandial hyperglycemia, which plays a crucial role in the development of diabetic conditions. In the current study, we have performed molecular docking of novel coumarin derivatives and proposed CD-59 as a potential dual inhibitor for the drug targets of interest, which were validated with MD simulations. Virtual screening using pharmacophore prediction identified four novel coumarin-like compounds ZINC02789441, ZINC40949448, ZINC13496808, and ZINC09781623. Out of these, ZINC02789441 has the most potential to act as a dual inhibitor against both the target enzymes, α-glucosidase, and α-amylase. However, both the novel coumarin derivatives, CD-59 and ZINC02789441 interact with the key residues of both the target enzymes, which in turn could alter the catalytic activity. This could reduce the biological enzymatic activity, which might result in the reduction of hyperglycemia. The docking results were supported by MD simulations and binding free energy calculations, where the binding of CD-59 and ZINC02789441 with the drug targets was validated. In addition, using the molecular scaffolds of the four potential dual inhibitors, more number f compounds can be designed and can be used for drug discovery. To conclude, the coumarin derivative CD-59 and the screened molecule ZINC02789441 could act as potential dual inhibitors against α-glucosidase and α-amylase for the management of post-prandial hyperglycemia.

## Figures and Tables

**Figure 1 molecules-27-03888-f001:**
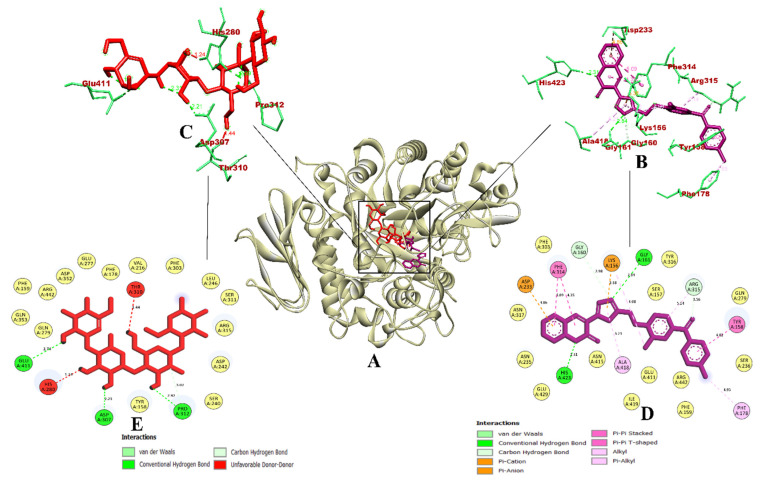
Visualization of binding interaction of CD-59 (violet) and acarbose (red) with α-glucosidase; (**A**) 3D structure of α-glucosidase with bound CD-59 and acarbose in the binding pocket, (**B**) 3D representation of CD-59 binding interactions, (**C**) 3D representation of acarbose binding interactions, and (**D**) 2D representation of CD-59 binding interactions, and (**E**) 2D representation of acarbose binding interactions. Colored: bound residues, yellow: surrounding residues.

**Figure 2 molecules-27-03888-f002:**
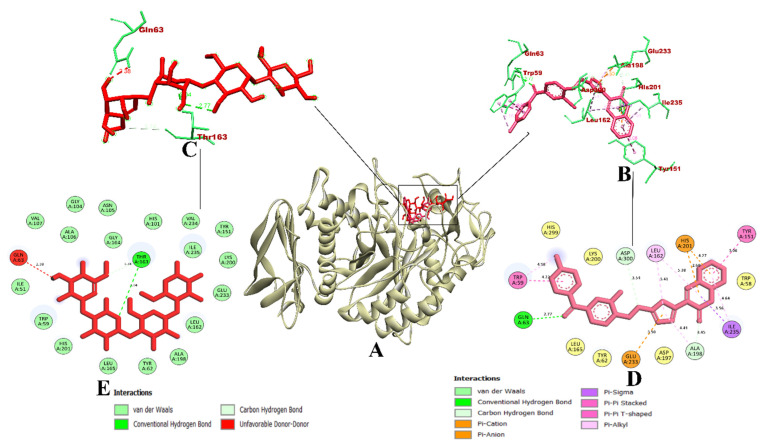
Visualization of binding interaction of CD-59 (magenta) and acarbose (red) with α-amylase; (**A**) 3D structure of α-amylase with bound CD-59 and acarbose in the binding pocket, (**B**) 3D representation of CD-59 binding interactions, (**C**) 3D representation of acarbose binding interactions, and (**D**) 2D representation of CD-59 binding interactions, and (**E**) 2D representation of acarbose binding interactions. Colored: bound residues, yellow: surrounding residues.

**Figure 3 molecules-27-03888-f003:**
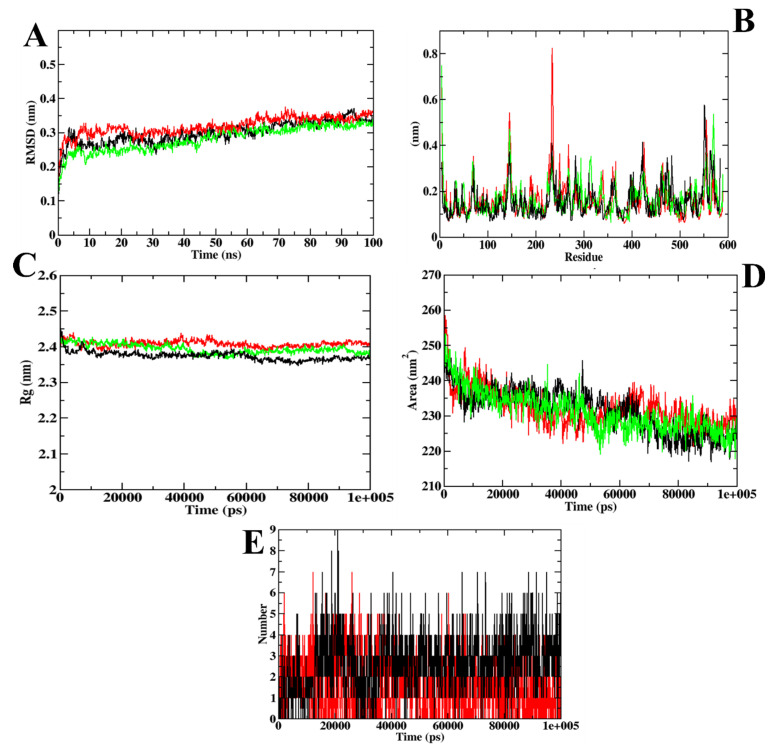
Visualization of MD trajectories obtained for CD-59 and acarbose complexed with α-glucosidase; (**A**) RMSD, (**B**) RMSF, (**C**) Rg, (**D**) SASA, and (**E**) SASA. Green: apoprotein, red: protein–acarbose complex, and black: protein–CD-59 complex.

**Figure 4 molecules-27-03888-f004:**
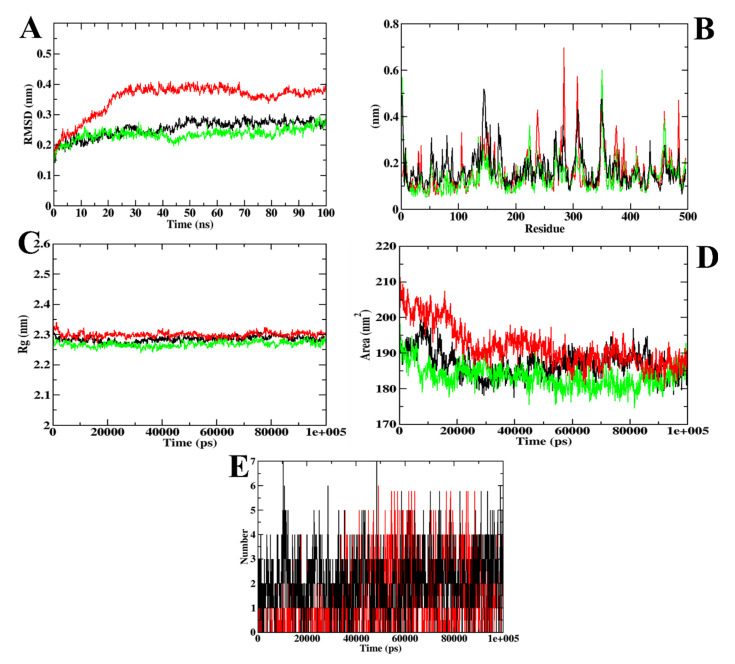
Visualization of MD trajectories obtained for CD-59 and acarbose complexed with α-amylase; (**A**) RMSD, (**B**) RMSF, (**C**) Rg, (**D**) SASA, and (**E**) SASA. Green: apoprotein, red: protein–acarbose complex, and black: protein–CD-59 complex.

**Figure 5 molecules-27-03888-f005:**
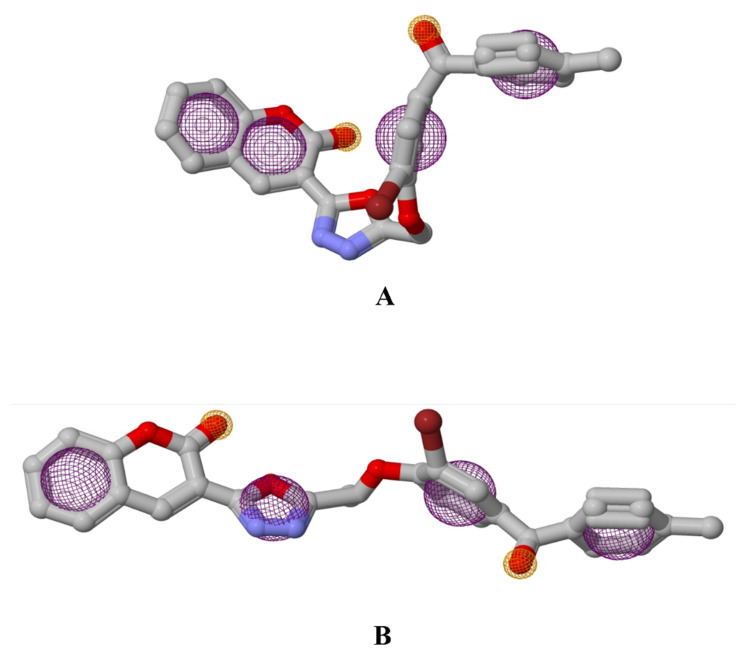
Pharmacophore models of CD-59 with (**A**) α-glucosidase and (**B**) α-amylase.

**Figure 6 molecules-27-03888-f006:**
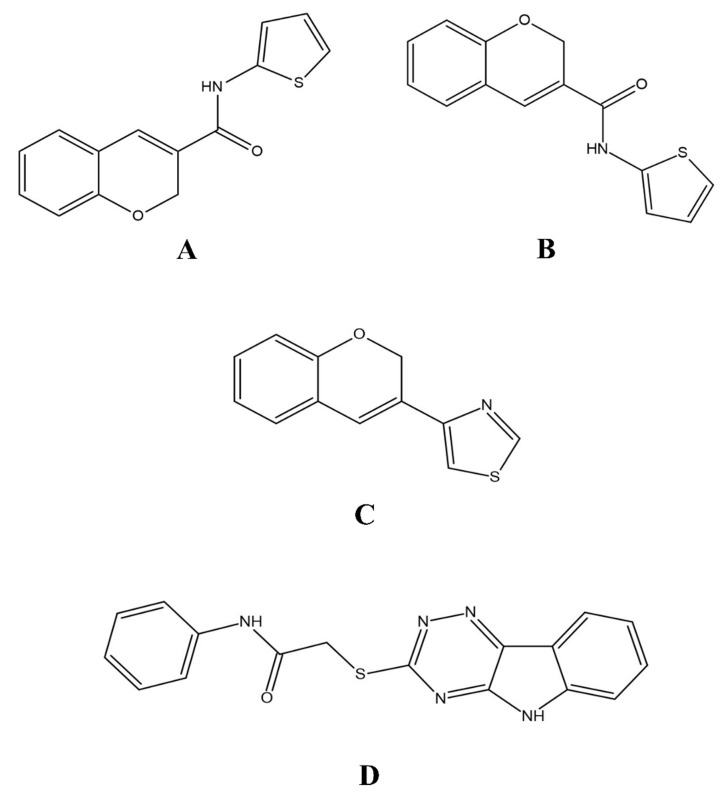
The 2D structures of the coumarin scaffolds obtained for the selected potential dual inhibitors from the ZINC database: (**A**) thiophenyl chromene carboxamide (scaffold of ZINC02789441), (**B**) thiophenyl chromene carboxamide with a different conformation (scaffold of ZINC40949448), (**C**) chromenyl thiazole (scaffold of ZINC09781623), and (**D**) triazino indol thio phenylacetamide (scaffold of ZINC13496808).

**Figure 7 molecules-27-03888-f007:**
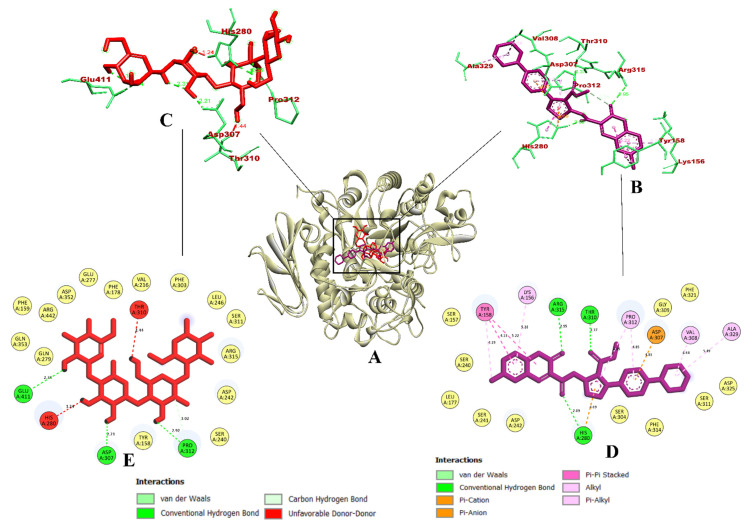
Visualization of binding interaction of ZINC02789441 (violet) and acarbose (red) with α-glucosidase; (**A**) 3D structure of α-glucosidase with bound ZINC02789441 and acarbose in the binding pocket, (**B**) 3D representation of ZINC02789441 binding interactions, (**C**) 3D representation of acarbose binding interactions, and (**D**) 2D representation of ZINC02789441 binding interactions, and (**E**) 2D representation of acarbose binding interactions. Colored: bound residues, yellow: surrounding residues.

**Figure 8 molecules-27-03888-f008:**
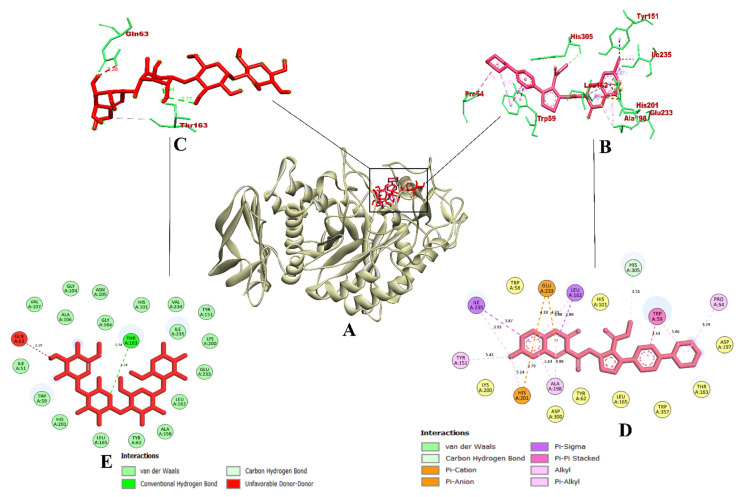
Visualization of binding interaction of ZINC02789441 (magenta) and acarbose (red) with α-amylase; (**A**) 3D structure of α-amylase with bound ZINC02789441 and acarbose in the binding pocket, (**B**) 3D representation of ZINC02789441 binding interactions, (**C**) 3D representation of acarbose binding interactions, and (**D**) 2D representation of CD-59 binding interactions, and (**E**) 2D representation of acarbose binding interactions. Colored: bound residues, yellow: surrounding residues.

**Figure 9 molecules-27-03888-f009:**
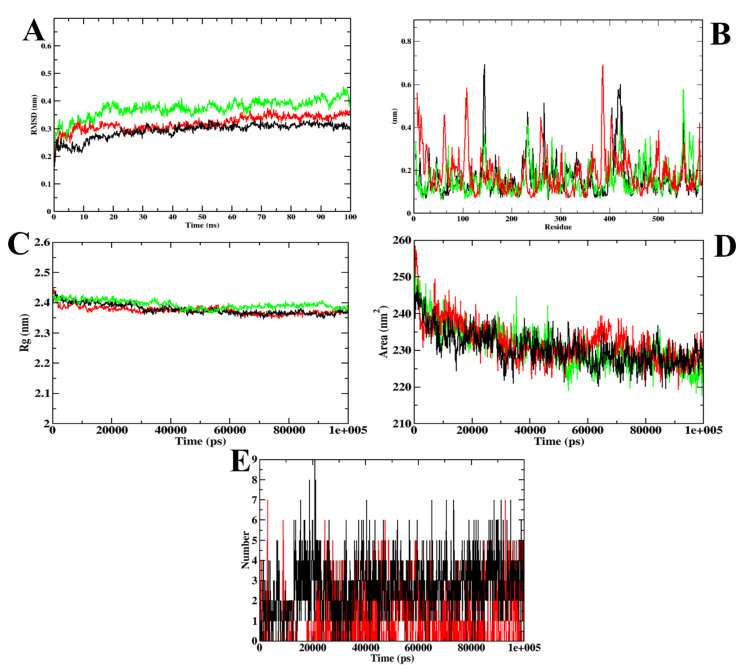
Visualization of MD trajectories obtained for ZINC02789441 and acarbose complexed with α-glucosidase; (**A**) RMSD, (**B**) RMSF, (**C**) Rg, (**D**) SASA, and (**E**) SASA. Green: apoprotein, red: protein–acarbose complex, and black: protein–ZINC02789441 complex.

**Figure 10 molecules-27-03888-f010:**
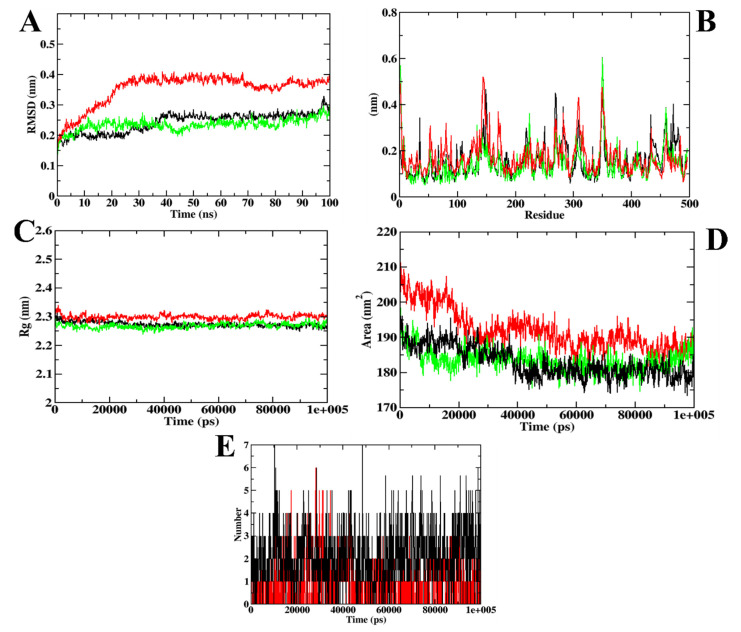
Visualization of MD trajectories obtained for ZINC02789441 and acarbose complexed with α-amylase; (**A**) RMSD, (**B**) RMSF, (**C**) Rg, (**D**) SASA, and (**E**) SASA. Green: apoprotein, red: protein–acarbose complex, and black: protein–ZINC02789441 complex.

**Table 1 molecules-27-03888-t001:** Structural details of coumarin derivatives used in the study.

Sl. No	Structure	Molecular Structural Differences
1	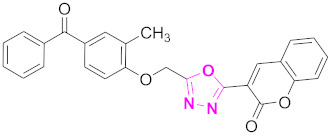 (4-((5-(2H-chromen-2-one)-1,3,4-oxadiazol-2-yl)methoxy)-3-methylphenyl)(phenyl)methanone (CD-1)	Basic skeleton comprises coumarin, and benzophenone bridged via 1,3,4-oxadiazole, Additional one methyl substituent present at ortho position of phenyl ring of benzophenone, no substituents at benzoyl ring as well as coumarin
2	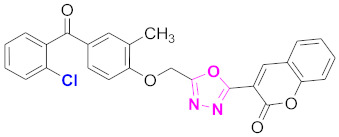 (4-((5-(2H-chromen-2-one)-1,3,4-oxadiazol-2-yl)methoxy)-3-methylphenyl)(2-chlorophenyl)methanone (CD-2)	With basic skeleton, chloro group is present at ortho position of benzoyl ring of benzophenone
3	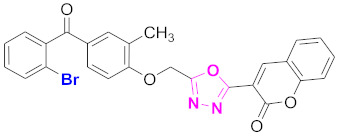 (4-((5-(2H-chromen-2-one)-1,3,4-oxadiazol-2-yl)methoxy)-3-methylphenyl)(2-bromophenyl)methanone (CD-3)	With basic skeleton, bromo group is present at ortho position of benzoyl ring of benzophenone
4	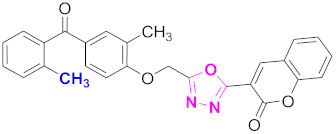 (4-((5-(2H-chromen-2-one)-1,3,4-oxadiazol-2-yl)methoxy)-3-methylphenyl)(2-methylphenyl)methanone (CD-4)	With basic skeleton, methyl group is present at ortho position of benzoyl ring of benzophenone
5	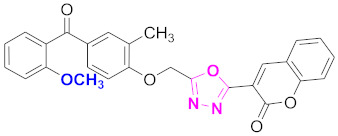 (4-((5-(2H-chromen-2-one)-1,3,4-oxadiazol-2-yl)methoxy)-3-methylphenyl)(2-methoxyphenyl)methanone (CD-5)	With basic skeleton, methoxy group is present at ortho position of benzoyl ring of benzophenone
6	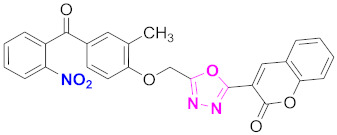 (4-((5-(2H-chromen-2-one)-1,3,4-oxadiazol-2-yl)methoxy)-3-methylphenyl)(2-nitrophenyl)methanone (CD-6)	With basic skeleton, nitro group is present at ortho position of benzoyl ring of benzophenone
7	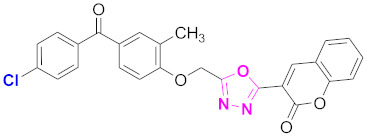 (4-((5-(2H-chromen-2-one)-1,3,4-oxadiazol-2-yl)methoxy)-3-methylphenyl)(4-chlorophenyl)methanone (CD-7)	With basic skeleton, chloro group is present at para position of benzoyl ring of benzophenone
8	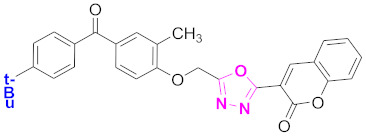 (4-((5-(2H-chromen-2-one)-1,3,4-oxadiazol-2-yl)methoxy)-3-methylphenyl)(4-t-butylphenyl)methanone (CD-8)	With basic skeleton, tertiary butyl group is present at para position of benzoyl ring of benzophenone
9	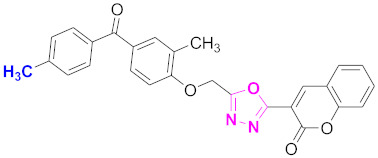 (4-((5-(2H-chromen-2-one)-1,3,4-oxadiazol-2-yl)methoxy)-3-methylphenyl)(4-methylphenyl)methanone (CD-9)	With basic skeleton, methyl group is present at para position of benzoyl ring of benzophenone
10	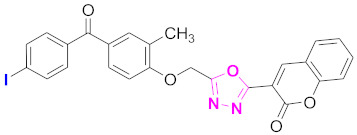 (4-((5-(2H-chromen-2-one)-1,3,4-oxadiazol-2-yl)methoxy)-3-methylphenyl)(4-iodophenyl)methanone (CD-10)	With basic skeleton, iodo group is present at para position of benzoyl ring of benzophenone
11	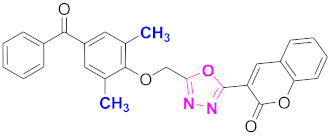 (4-((5-(2H-chromen-2-one)-1,3,4-oxadiazol-2-yl)methoxy)-3,5-dimethylphenyl)(phenyl)methanone (CD-11)	With basic skeleton, 2-methyl groups present at ortho position of phenyl ring of benzophenone, no substitutions at benzoyl ring of benzophenone
12	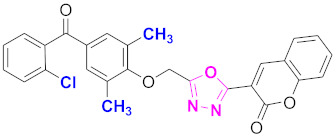 (4-((5-(2H-chromen-2-one)-1,3,4-oxadiazol-2-yl)methoxy)-3,5-dimethylphenyl)(2-chlorophenyl)methanone (CD-12)	With basic skeleton, chloro group is present at ortho position of benzoyl ring of benzophenone
13	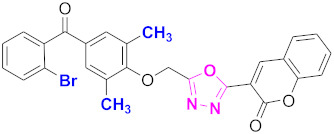 (4-((5-(2H-chromen-2-one)-1,3,4-oxadiazol-2-yl)methoxy)-3,5-dimethylphenyl)(2-bromophenyl)methanone (CD-13)	With basic skeleton, bromo group is present at ortho position of benzoyl ring of benzophenone
14	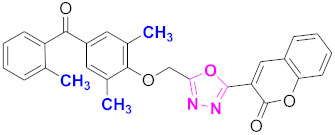 (4-((5-(2H-chromen-2-one)-1,3,4-oxadiazol-2-yl)methoxy)-3,5-dimethylphenyl)(2-methylphenyl)methanone (CD-14)	With basic skeleton, methyl group is present at ortho position of benzoyl ring of benzophenone
15	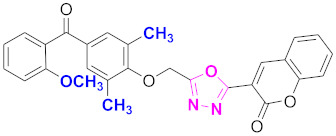 (4-((5-(2H-chromen-2-one)-1,3,4-oxadiazol-2-yl)methoxy)-3,5-dimethylphenyl)(2-methoxyphenyl)methanone (CD-15)	With basic skeleton, methoxy group is present at ortho position of benzoyl ring of benzophenone
16	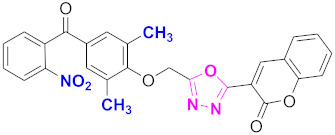 (4-((5-(2H-chromen-2-one)-1,3,4-oxadiazol-2-yl)methoxy)-3,5-dimethylphenyl)(2-nitrophenyl)methanone (CD-16)	With basic skeleton, nitro group is present at ortho position of benzoyl ring of benzophenone
17	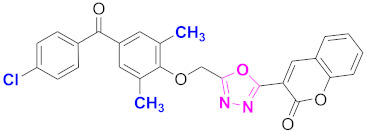 (4-((5-(2H-chromen-2-one)-1,3,4-oxadiazol-2-yl)methoxy)-3,5-dimethylphenyl)(4-chlorophenyl)methanone (CD-17)	With basic skeleton, chloro group is present at para position of benzoyl ring of benzophenone
18	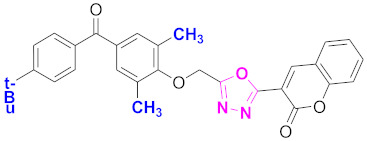 (4-((5-(2H-chromen-2-one)-1,3,4-oxadiazol-2-yl)methoxy)-3,5-dimethylphenyl)(4-t-butylphenyl)methanone (CD-18)	With basic skeleton, t-butyl group is present at para position of benzoyl ring of benzophenone
19	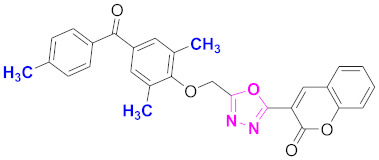 (4-((5-(2H-chromen-2-one)-1,3,4-oxadiazol-2-yl)methoxy)-3,5-dimethylphenyl)(4-methylphenyl)methanone (CD-19)	With basic skeleton, methyl group is present at para position of benzoyl ring of benzophenone
20	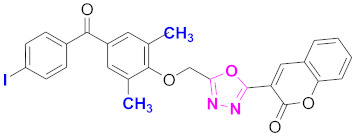 (4-((5-(2H-chromen-2-one)-1,3,4-oxadiazol-2-yl)methoxy)-3,5-dimethylphenyl)(4-iodophenyl)methanone (CD-20)	With basic skeleton, iodo group is present at para position of benzoyl ring of benzophenone
21	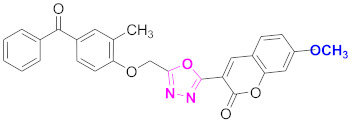 (4-((5-(7-methoxy-2H-chromen-2-one)-1,3,4-oxadiazol-2-yl)methoxy)-3-methylphenyl)(phenyl)methanone (CD-21)	With basic skeleton, coumarin contains methoxy group at 7th position, methyl group is present at ortho position of phenyl ring of benzophenone
22	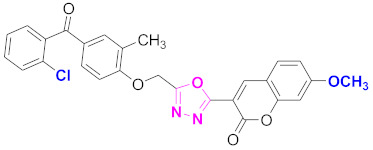 4-((5-(7-methoxy-2H-chromen-2-one)-1,3,4-oxadiazol-2-yl)methoxy)-3-methylphenyl)(2-chlorophenyl)methanone (CD-22)	With basic skeleton, coumarin contains methoxy group at 7th position, methyl group is present at ortho position of phenyl ring of benzophenone and chloro group is present at ortho position of benzoyl ring of benzophenone
23	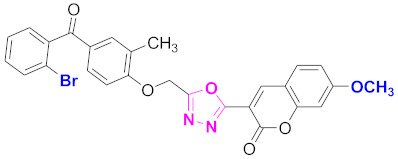 4-((5-(7-methoxy-2H-chromen-2-one)-1,3,4-oxadiazol-2-yl)methoxy)-3-methylphenyl)(2-bromophenyl)methanone (CD-23)	With basic skeleton, coumarin contains methoxy group at 7th position, methyl group is present at ortho position of phenyl ring of benzophenone and bromo group is present at ortho position of benzoyl ring of benzophenone
24	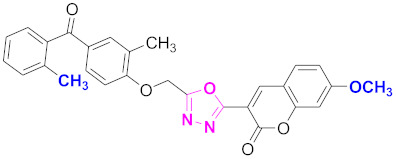 4-((5-(7-methoxy-2H-chromen-2-one)-1,3,4-oxadiazol-2-yl)methoxy)-3-methylphenyl)(2-methylphenyl)methanone (CD-24)	With basic skeleton, coumarin contains methoxy group at 7th position, methyl group is present at ortho position of phenyl ring of benzophenone and methyl group is present at ortho position of benzoyl ring of benzophenone
25	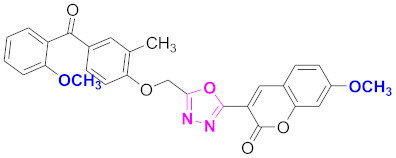 4-((5-(7-methoxy-2H-chromen-2-one)-1,3,4-oxadiazol-2-yl)methoxy)-3-methylphenyl)(2-methoxyphenyl)methanone (CD-25)	With basic skeleton, coumarin contains methoxy group at 7th position, methyl group is present at ortho position of phenyl ring of benzophenone and methoxy group is present at ortho position of benzoyl ring of benzophenone
26	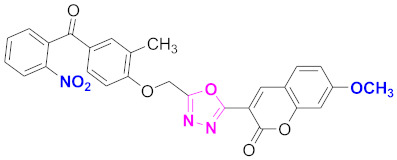 4-((5-(7-methoxy-2H-chromen-2-one)-1,3,4-oxadiazol-2-yl)methoxy)-3-methylphenyl)(2-nitrophenyl)methanone (CD-26)	With basic skeleton, coumarin contains methoxy group at 7th position, methyl group is present at ortho position of phenyl ring of benzophenone and nitro group is present at ortho position of benzoyl ring of benzophenone
27	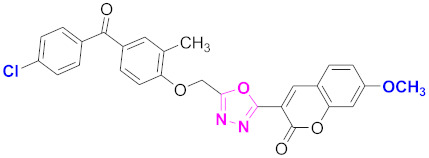 4-((5-(7-methoxy-2H-chromen-2-one)-1,3,4-oxadiazol-2-yl)methoxy)-3-methylphenyl)(4-chlorophenyl)methanone (CD-27)	With basic skeleton, coumarin contains methoxy group at 7th position, methyl group is present at ortho position of phenyl ring of benzophenone and chloro group is present at para position of benzoyl ring of benzophenone
28	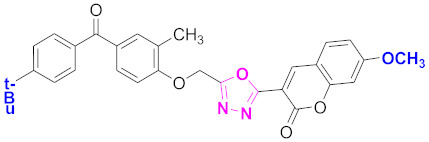 4-((5-(7-methoxy-2H-chromen-2-one)-1,3,4-oxadiazol-2-yl)methoxy)-3-methylphenyl)(4-t-butylphenyl)methanone (CD-28)	With basic skeleton, coumarin contains methoxy group at 7th position, methyl group is present at ortho position of phenyl ring of benzophenone and t-butyl group is present at para position of benzoyl ring of benzophenone
29	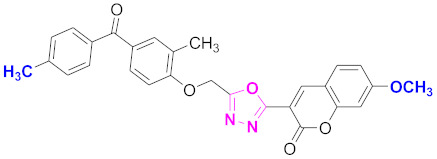 4-((5-(7-methoxy-2H-chromen-2-one)-1,3,4-oxadiazol-2-yl)methoxy)-3-methylphenyl)(4-methylphenyl)methanone (CD-29)	With basic skeleton, coumarin contains methoxy group at 7th position, methyl group is present at ortho position of phenyl ring of benzophenone and methyl group is present at para position of benzoyl ring of benzophenone
30	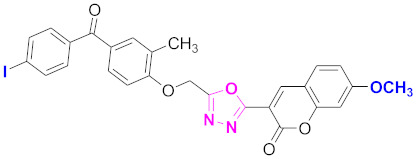 4-((5-(7-methoxy-2H-chromen-2-one)-1,3,4-oxadiazol-2-yl)methoxy)-3-methylphenyl)(4-iodophenyl)methanone (CD-30)	With basic skeleton, coumarin contains methoxy group at 7th position, methyl group is present at ortho position of phenyl ring of benzophenone and iodo group is present at para position of benzoyl ring of benzophenone
31	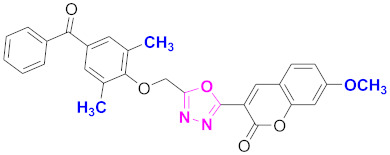 (4-((5-(7-methoxy-2H-chromen-2-one)-1,3,4-oxadiazol-2-yl)methoxy)-3,5-dimethylphenyl)(phenyl)methanone (CD-31)	With basic skeleton, coumarin contains methoxy group at 7th position, 2-methyl groups present at ortho position of phenyl ring of benzophenone, and no substitutions at benzoyl ring of benzophenone
32	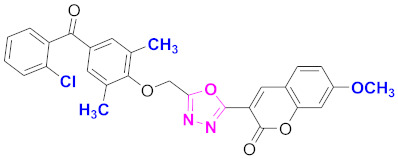 4-((5-(7-methoxy-2H-chromen-2-one)-1,3,4-oxadiazol-2-yl)methoxy)-3,5-dimethylphenyl)(2-chlorophenyl)methanone (CD-32)	With basic skeleton, coumarin contains methoxy group at 7th position, 2-methyl groups present at ortho position of phenyl ring of benzophenone and chloro group is present at ortho position of benzoyl ring of benzophenone
33	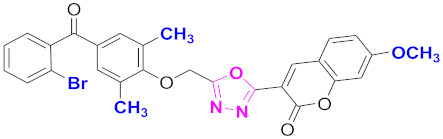 4-((5-(7-methoxy-2H-chromen-2-one)-1,3,4-oxadiazol-2-yl)methoxy)-3,5-dimethylphenyl)(2-bromophenyl)methanone (CD-33)	With basic skeleton, coumarin contains methoxy group at 7th position, 2-methyl groups present at ortho position of phenyl ring of benzophenone and bromo group is present at ortho position of benzoyl ring of benzophenone
34	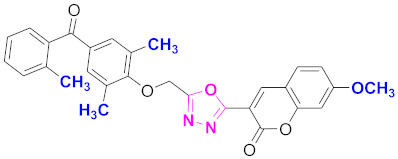 4-((5-(7-methoxy-2H-chromen-2-one)-1,3,4-oxadiazol-2-yl)methoxy)-3,5-dimethylphenyl)(2-methylphenyl)methanone (CD-34)	With basic skeleton, coumarin contains methoxy group at 7th position, 2-methyl groups present at ortho position of phenyl ring of benzophenone and methyl group is present at ortho position of benzoyl ring of benzophenone
35	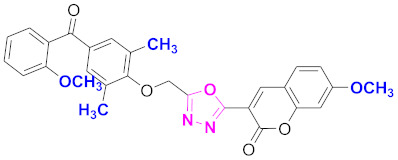 4-((5-(7-methoxy-2H-chromen-2-one)-1,3,4-oxadiazol-2-yl)methoxy)-3,5-dimethylphenyl)(2-methoxyphenyl)methanone (CD-35)	With basic skeleton, coumarin contains methoxy group at 7th position, 2-methyl groups present at ortho position of phenyl ring of benzophenone and methoxy group is present at ortho position of benzoyl ring of benzophenone
36	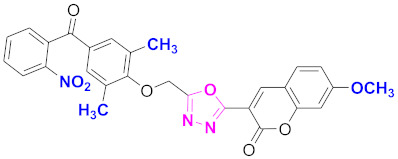 4-((5-(7-methoxy-2H-chromen-2-one)-1,3,4-oxadiazol-2-yl)methoxy)-3,5-dimethylphenyl)(2-nitrophenyl)methanone (CD-36)	With basic skeleton, coumarin contains methoxy group at 7th position, 2-methyl groups present at ortho position of phenyl ring of benzophenone and nitro group is present at ortho position of benzoyl ring of benzophenone
37	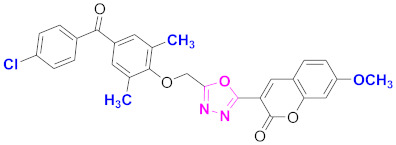 4-((5-(7-methoxy-2H-chromen-2-one)-1,3,4-oxadiazol-2-yl)methoxy)-3,5-dimethylphenyl)(4-chlorophenyl)methanone (CD-37)	With basic skeleton, coumarin contains methoxy group at 7th position, 2-methyl groups present at ortho position of phenyl ring of benzophenone and chloro group is present at para position of benzoyl ring of benzophenone
38	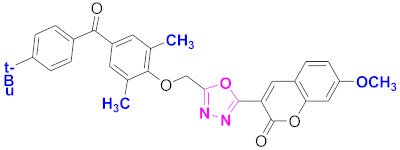 4-((5-(7-methoxy-2H-chromen-2-one)-1,3,4-oxadiazol-2-yl)methoxy)-3,5-dimethylphenyl)(4-t-butylphenyl)methanone (CD-38)	With basic skeleton, coumarin contains methoxy group at 7th position, 2-methyl groups present at ortho position of phenyl ring of benzophenone and t-butyl group is present at para position of benzoyl ring of benzophenone
39	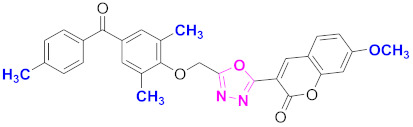 4-((5-(7-methoxy-2H-chromen-2-one)-1,3,4-oxadiazol-2-yl)methoxy)-3,5-dimethylphenyl)(4-methylphenyl)methanone (CD-39)	With basic skeleton, coumarin contains methoxy group at 7th position, 2-methyl groups present at ortho position of phenyl ring of benzophenone and methyl group is present at para position of benzoyl ring of benzophenone
40	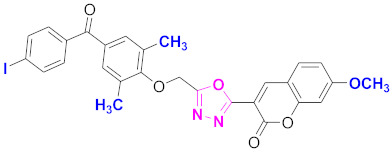 4-((5-(7-methoxy-2H-chromen-2-one)-1,3,4-oxadiazol-2-yl)methoxy)-3,5-dimethylphenyl)(4-iodophenyl)methanone (CD-40)	With basic skeleton, coumarin contains methoxy group at 7th position, 2-methyl groups present at ortho position of phenyl ring of benzophenone and iodo group is present at para position of benzoyl ring of benzophenone
41	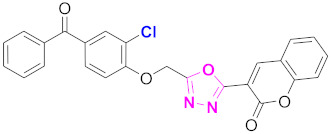 (4-((5-(2H-chromen-2-one)-1,3,4-oxadiazol-2-yl)methoxy)-3-chlorolphenyl)(phenyl)methanone (CD-41)	Basic skeleton comprises coumarin, and benzophenone bridged via 1,3,4-oxadiazole, Additional one chloro substituent present at ortho position of phenyl ring of benzophenone, no substituents at benzoyl ring as well as coumarin
42	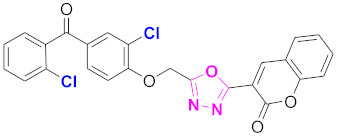 (4-((5-(2H-chromen-2-one)-1,3,4-oxadiazol-2-yl)methoxy)-3-chlorophenyl)(2-chlorophenyl)methanone (CD-42)	With basic skeleton, additional one chloro substituent present at ortho position of phenyl ring of benzophenone, and chloro group is present at ortho position of benzoyl ring of benzophenone
43	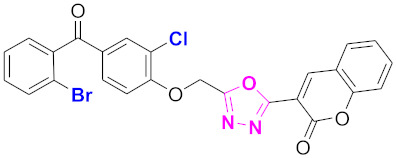 (4-((5-(2H-chromen-2-one)-1,3,4-oxadiazol-2-yl)methoxy)-3-chlorophenyl)(2-bromophenyl)methanone (CD-43)	With basic skeleton, additional one chloro substituent present at ortho position of phenyl ring of benzophenone, and bromo group is present at ortho position of benzoyl ring of benzophenone
44	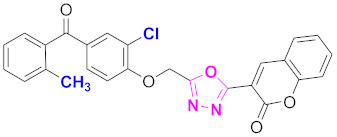 (4-((5-(2H-chromen-2-one)-1,3,4-oxadiazol-2-yl)methoxy)-3-chlorophenyl)(2-methylphenyl)methanone (CD-44)	With basic skeleton, additional one chloro substituent present at ortho position of phenyl ring of benzophenone, and methyl group is present at ortho position of benzoyl ring of benzophenone
45	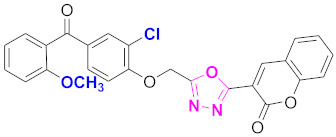 (4-((5-(2H-chromen-2-one)-1,3,4-oxadiazol-2-yl)methoxy)-3-chlorophenyl)(2-methoxyphenyl)methanone (CD-45)	With basic skeleton, additional one chloro substituent present at ortho position of phenyl ring of benzophenone, and methoxy group is present at ortho position of benzoyl ring of benzophenone
46	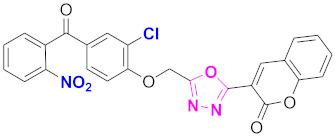 (4-((5-(2H-chromen-2-one)-1,3,4-oxadiazol-2-yl)methoxy)-3-chlorophenyl)(2-nitrophenyl)methanone (CD-46)	With basic skeleton, additional one chloro substituent present at ortho position of phenyl ring of benzophenone, and nitro group is present at ortho position of benzoyl ring of benzophenone
47	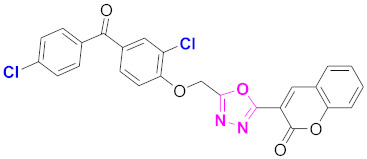 (4-((5-(2H-chromen-2-one)-1,3,4-oxadiazol-2-yl)methoxy)-3-chlorophenyl)(4-chlorophenyl)methanone (CD-47)	With basic skeleton, additional one chloro substituent present at ortho position of phenyl ring of benzophenone, and chloro group is present at para position of benzoyl ring of benzophenone
48	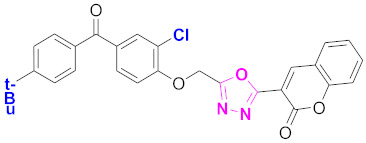 (4-((5-(2H-chromen-2-one)-1,3,4-oxadiazol-2-yl)methoxy)-3-chlorophenyl)(4-t-butylphenyl)methanone (CD-48)	With basic skeleton, additional one chloro substituent present at ortho position of phenyl ring of benzophenone, and t-butyl group is present at para position of benzoyl ring of benzophenone
49	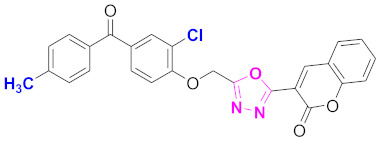 (4-((5-(2H-chromen-2-one)-1,3,4-oxadiazol-2-yl)methoxy)-3-chlorophenyl)(4-methylphenyl)methanone (CD-49)	With basic skeleton, additional one chloro substituent present at ortho position of phenyl ring of benzophenone, and methyl group is present at para position of benzoyl ring of benzophenone
50	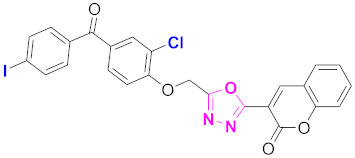 (4-((5-(2H-chromen-2-one)-1,3,4-oxadiazol-2-yl)methoxy)-3-chlorophenyl)(4-iodophenyl)methanone (CD-50)	With basic skeleton, additional one chloro substituent present at ortho position of phenyl ring of benzophenone, and iodo group is present at para position of benzoyl ring of benzophenone
51	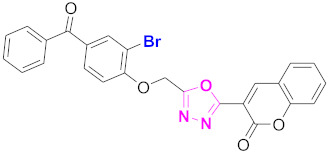 (4-((5-(2H-chromen-2-one)-1,3,4-oxadiazol-2-yl)methoxy)-3-bromophenyl)(phenyl)methanone (CD-51)	Basic skeleton comprises coumarin, and benzophenone bridged via 1,3,4-oxadiazole, Additional one bromo group is present at ortho position of phenyl ring of benzophenone, no substituents at benzoyl ring as well as coumarin
52	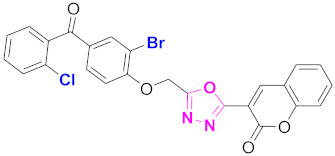 (4-((5-(2H-chromen-2-one)-1,3,4-oxadiazol-2-yl)methoxy)-3-bromophenyl)(2-chlorophenyl)methanone (CD-52)	With basic skeleton, additional one bromo substituent present at ortho position of phenyl ring of benzophenone, and chloro group is present at ortho position of benzoyl ring of benzophenone
53	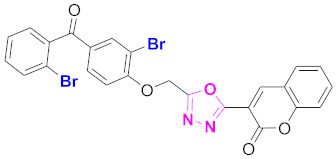 (4-((5-(2H-chromen-2-one)-1,3,4-oxadiazol-2-yl)methoxy)-3-bromophenyl)(2-bromophenyl)methanone (CD-53)	With basic skeleton, additional one bromo substituent present at ortho position of phenyl ring of benzophenone, and bromo group is present at ortho position of benzoyl ring of benzophenone
54	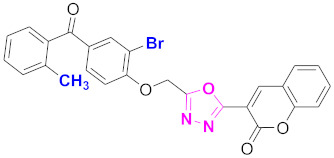 (4-((5-(2H-chromen-2-one)-1,3,4-oxadiazol-2-yl)methoxy)-3-bromophenyl)(2-methylphenyl)methanone (CD-54)	With basic skeleton, additional one bromo substituent present at ortho position of phenyl ring of benzophenone, and methyl group is present at ortho position of benzoyl ring of benzophenone
55	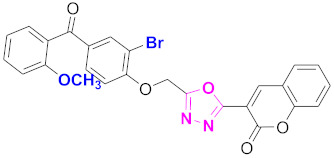 (4-((5-(2H-chromen-2-one)-1,3,4-oxadiazol-2-yl)methoxy)-3-bromophenyl)(2-methoxyphenyl)methanone (CD-55)	With basic skeleton, additional one bromo substituent present at ortho position of phenyl ring of benzophenone, and methoxy group is present at ortho position of benzoyl ring of benzophenone
56	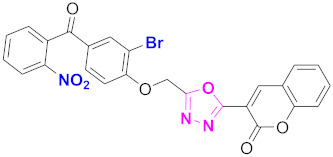 (4-((5-(2H-chromen-2-one)-1,3,4-oxadiazol-2-yl)methoxy)-3-bromophenyl)(2-nitrophenyl)methanone (CD-56)	With basic skeleton, additional one bromo substituent present at ortho position of phenyl ring of benzophenone, and nitro group is present at ortho position of benzoyl ring of benzophenone
57	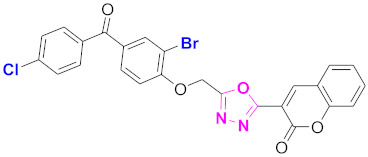 (4-((5-(2H-chromen-2-one)-1,3,4-oxadiazol-2-yl)methoxy)-3-bromophenyl)(4-chlorophenyl)methanone (CD-57)	With basic skeleton, additional one bromo substituent present at ortho position of phenyl ring of benzophenone, and chloro group is present at para position of benzoyl ring of benzophenone
58	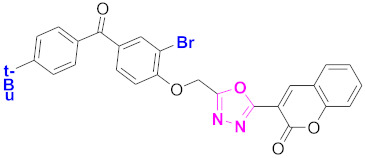 (4-((5-(2H-chromen-2-one)-1,3,4-oxadiazol-2-yl)methoxy)-3-bromophenyl)(4-t-butylphenyl)methanone (CD-58)	With basic skeleton, additional one bromo substituent present at ortho position of phenyl ring of benzophenone, and t-butyl group is present at para position of benzoyl ring of benzophenone
59	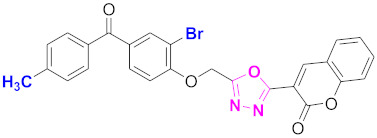 (4-((5-(2H-chromen-2-one)-1,3,4-oxadiazol-2-yl)methoxy)-3-bromophenyl)(4-methylphenyl)methanone (CD-59)	With basic skeleton, additional one bromo substituent present at ortho position of phenyl ring of benzophenone, and methyl group is present at para position of benzoyl ring of benzophenone
60	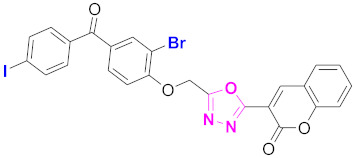 (4-((5-(2H-chromen-2-one)-1,3,4-oxadiazol-2-yl)methoxy)-3-bromophenyl)(4-iodophenyl)methanone (CD-60)	With basic skeleton, additional one bromo substituent present at ortho position of phenyl ring of benzophenone, and iodo group is present at para position of benzoyl ring of benzophenone
61	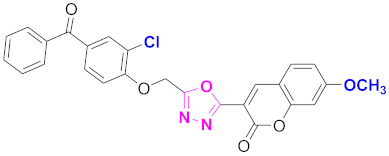 (4-((5-(7-methoxy-2H-chromen-2-one)-1,3,4-oxadiazol-2-yl)methoxy)-3-chlorophenyl)(phenyl)methanone (CD-61)	With basic skeleton, coumarin contains methoxy group at 7th position, and chloro group is present at ortho position of phenyl ring of benzophenone and no substitutions at benzoyl ring of benzophenone
62	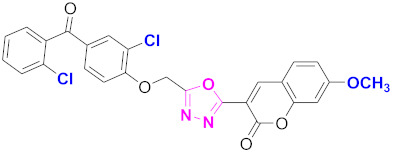 4-((5-(7-methoxy-2H-chromen-2-one)-1,3,4-oxadiazol-2-yl)methoxy)-3-chlorophenyl)(2-chlorophenyl)methanone (CD-62)	With basic skeleton, coumarin contains methoxy group at 7th position, and chloro group is present at ortho position of phenyl ring of benzophenone and chloro group is present at ortho position of benzoyl ring of benzophenone
63	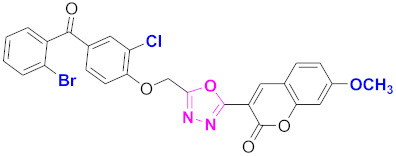 4-((5-(7-methoxy-2H-chromen-2-one)-1,3,4-oxadiazol-2-yl)methoxy)-3-chlorophenyl)(2-bromophenyl)methanone (CD-63)	With basic skeleton, coumarin contains methoxy group at 7th position, and chloro group is present at ortho position of phenyl ring of benzophenone and bromo group is present at ortho position of benzoyl ring of benzophenone
64	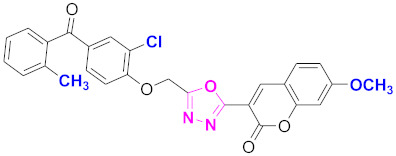 4-((5-(7-methoxy-2H-chromen-2-one)-1,3,4-oxadiazol-2-yl)methoxy)-3-chlorophenyl)(2-methylphenyl)methanone (CD-64)	With basic skeleton, coumarin contains methoxy group at 7th position, and chloro group is present at ortho position of phenyl ring of benzophenone, and methyl group is present at ortho position of benzoyl ring of benzophenone
65	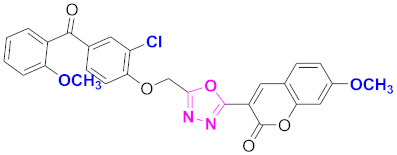 4-((5-(7-methoxy-2H-chromen-2-one)-1,3,4-oxadiazol-2-yl)methoxy)-3-chlorophenyl)(2-methoxyphenyl)methanone (CD-65)	With basic skeleton, coumarin contains methoxy group at 7th position, and chloro group is present at ortho position of phenyl ring of benzophenone and methoxy group is present at ortho position of benzoyl ring of benzophenone
66	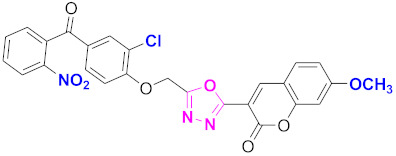 4-((5-(7-methoxy-2H-chromen-2-one)-1,3,4-oxadiazol-2-yl)methoxy)-3-chlorophenyl)(2-nitrophenyl)methanone (CD-66)	With basic skeleton, coumarin contains methoxy group at 7th position, and chloro group is present at ortho position of phenyl ring of benzophenone, and nitro group is present at ortho position of benzoyl ring of benzophenone
67	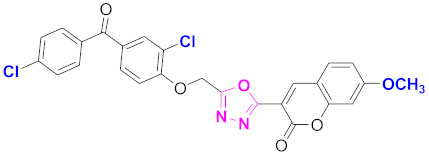 4-((5-(7-methoxy-2H-chromen-2-one)-1,3,4-oxadiazol-2-yl)methoxy)-3-chlorophenyl)(4-chlorophenyl)methanone (CD-67)	With basic skeleton, coumarin contains methoxy group at 7th position, and chloro group is present at ortho position of phenyl ring of benzophenone and chloro group is present at para position of benzoyl ring of benzophenone
68	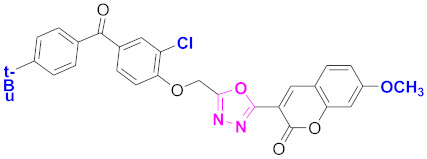 4-((5-(7-methoxy-2H-chromen-2-one)-1,3,4-oxadiazol-2-yl)methoxy)-3-chlorophenyl)(4-t-butylphenyl)methanone (CD-68)	With basic skeleton, coumarin contains methoxy group at 7th position, and chloro group is present at ortho position of phenyl ring of benzophenone and t-butyl group is present at para position of benzoyl ring of benzophenone
69	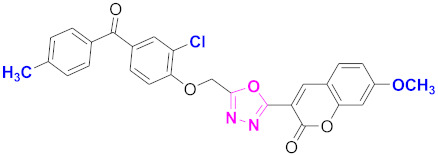 4-((5-(7-methoxy-2H-chromen-2-one)-1,3,4-oxadiazol-2-yl)methoxy)-3-chlorophenyl)(4-methylphenyl)methanone (CD-69)	With basic skeleton, coumarin contains methoxy group at 7th position, and chloro group is present at ortho position of phenyl ring of benzophenone, and methyl group is present at para position of benzoyl ring of benzophenone
70	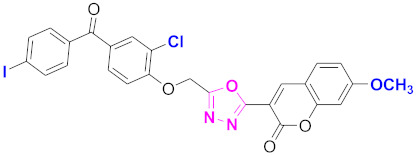 4-((5-(7-methoxy-2H-chromen-2-one)-1,3,4-oxadiazol-2-yl)methoxy)-3-chlorophenyl)(4-iodophenyl)methanone (CD-70)	With basic skeleton, coumarin contains methoxy group at 7th position, and chloro group is present at ortho position of phenyl ring of benzophenone and iodo group is present at para position of benzoyl ring of benzophenone
71	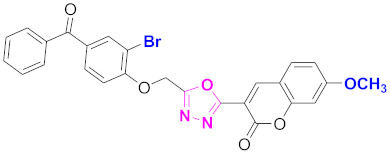 4-((5-(7-methoxy-2H-chromen-2-one)-1,3,4-oxadiazol-2-yl)methoxy)-3-bromophenyl)(phenyl)methanone (CD-71)	With basic skeleton, coumarin contains methoxy group at 7th position, and bromo group is present at ortho position of phenyl ring of benzophenone and no substitutions at benzoyl ring of benzophenone
72	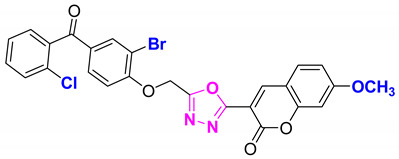 4-((5-(7-methoxy-2H-chromen-2-one)-1,3,4-oxadiazol-2-yl)methoxy)-3-bromophenyl)(2-chlorophenyl)methanone (CD-72)	With basic skeleton, coumarin contains methoxy group at 7th position, and bromo group is present at ortho position of phenyl ring of benzophenone and chloro group is present at ortho position of benzoyl ring of benzophenone
73	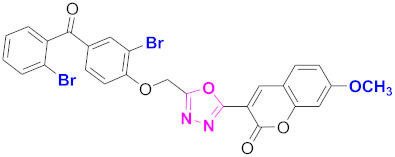 4-((5-(7-methoxy-2H-chromen-2-one)-1,3,4-oxadiazol-2-yl)methoxy)-3-bromophenyl)(2-bromophenyl)methanone (CD-73)	With basic skeleton, coumarin contains methoxy group at 7th position, and bromo group is present at ortho position of phenyl ring of benzophenone and bromo group is present at ortho position of benzoyl ring of benzophenone
74	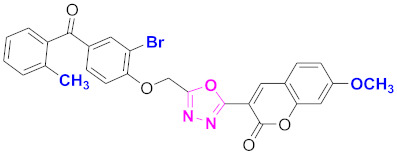 4-((5-(7-methoxy-2H-chromen-2-one)-1,3,4-oxadiazol-2-yl)methoxy)-3-bromophenyl)(2-methylphenyl)methanone (CD-74)	With basic skeleton, coumarin contains methoxy group at 7th position, and bromo group is present at ortho position of phenyl ring of benzophenone, and methyl group is present at ortho position of benzoyl ring of benzophenone
75	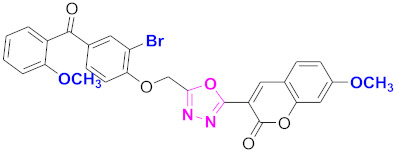 4-((5-(7-methoxy-2H-chromen-2-one)-1,3,4-oxadiazol-2-yl)methoxy)-3-bromophenyl)(2-methoxyphenyl)methanone (CD-75)	With basic skeleton, coumarin contains methoxy group at 7th position, and bromo group is present at ortho position of phenyl ring of benzophenone and methoxy group is present at ortho position of benzoyl ring of benzophenone
76	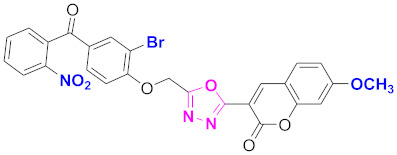 4-((5-(7-methoxy-2H-chromen-2-one)-1,3,4-oxadiazol-2-yl)methoxy)-3-bromophenyl)(2-nitrophenyl)methanone (CD-76)	With basic skeleton, coumarin contains methoxy group at 7th position, and bromo group is present at ortho position of phenyl ring of benzophenone, and nitro group is present at ortho position of benzoyl ring of benzophenone
77	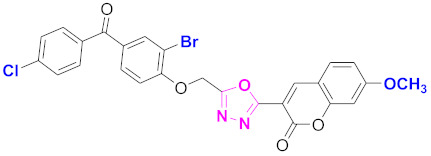 4-((5-(7-methoxy-2H-chromen-2-one)-1,3,4-oxadiazol-2-yl)methoxy)-3-bromophenyl)(4-chlorophenyl)methanone (CD-77)	With basic skeleton, coumarin contains methoxy group at 7th position, and bromo group is present at ortho position of phenyl ring of benzophenone and chloro group is present at para position of benzoyl ring of benzophenone
78	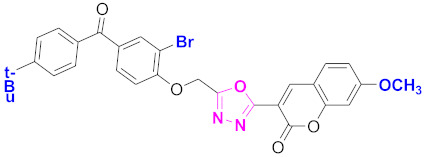 4-((5-(7-methoxy-2H-chromen-2-one)-1,3,4-oxadiazol-2-yl)methoxy)-3-bromophenyl)(4-t-butylphenyl)methanone (CD-78)	With basic skeleton, coumarin contains methoxy group at 7th position, and bromo group is present at ortho position of phenyl ring of benzophenone and t-butyl group is present at para position of benzoyl ring of benzophenone
79	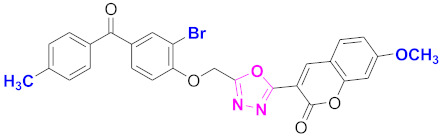 4-((5-(7-methoxy-2H-chromen-2-one)-1,3,4-oxadiazol-2-yl)methoxy)-3-bromophenyl)(4-methylphenyl)methanone (CD-79)	With basic skeleton, coumarin contains methoxy group at 7th position, and bromo group is present at ortho position of phenyl ring of benzophenone, and methyl group is present at para position of benzoyl ring of benzophenone
80	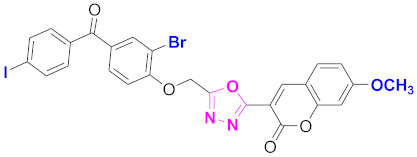 4-((5-(7-methoxy-2H-chromen-2-one)-1,3,4-oxadiazol-2-yl)methoxy)-3-bromophenyl)(4-iodophenyl)methanone (CD-80)	With basic skeleton, coumarin contains methoxy group at 7th position, and bromo group is present at ortho position of phenyl ring of benzophenone, and iodo group is present at para position of benzoyl ring of benzophenone

**Table 2 molecules-27-03888-t002:** Virtual screening of coumarin derivatives against α-glucosidase (PDB ID: 3A4A) and α-amylase (PDB ID: 2QV4).

Coumarin Derivative	Binding Affinity (kcal/mol)		Total No. of Intermolecular Interactions		Total No. of Hydrogen Bonds	
	α-Glucosidase	α-Amylase	α-Glucosidase	α-Amylase	α-Glucosidase	α-Amylase
58	−11.9	−11.1	16	16	5	2
78	−11.8	−10.7	18	14	6	2
68	−11.8	−10.6	16	17	6	1
18	−11.8	−11.3	10	16	2	1
28	−11.7	−10.7	11	20	4	2
38	−11.7	−10.8	10	17	2	3
59	−11.6	−11.3	14	16	4	3
8	−11.6	−11.2	11	15	2	2
48	−11.5	−11.1	11	14	2	1
29	−11.4	−10.7	14	13	5	1
11	−11.4	−11	13	12	2	1
19	−11.4	−11.4	11	16	2	1
49	−11.3	−11.3	16	17	3	2
47	−11.3	−11.1	14	17	4	3
67	−11.3	−10.9	14	16	5	2
10	−11.3	−11.2	13	13	4	1
27	−11.3	−10.9	13	13	3	1
69	−11.3	−11	12	14	3	1
62	−11.2	−10.7	14	11	4	1
74	−11.2	−10.7	14	13	4	3
77	−11.2	−10.5	14	14	3	3
79	−11.2	−10.5	14	15	3	1
22	−11.2	−10.8	13	11	5	1
64	−11.2	−10.7	13	11	4	2
24	−11.2	−10.5	11	17	2	1
1	−11.2	−11.3	9	14	1	1
32	−11.1	−11	16	11	6	1
33	−11.1	−11	16	12	5	1
80	−11.1	−10.8	16	15	5	2
70	−11.1	−10.9	15	14	4	1
34	−11.1	−11	14	9	6	1
39	−11.1	−10.7	14	14	2	3
72	−11.1	−10.6	13	13	3	2
76	−11.1	−10.2	13	18	5	4
26	−11.1	−10.4	12	13	4	4
41	−11.1	−10.9	11	15	2	2
9	−11.1	−10.7	11	9	2	1
30	−11.1	−11	10	14	3	1
2	−11.1	−10.9	10	10	4	1
42	−11.1	−11	9	10	4	3
14	−11.1	−10.6	9	10	1	-
66	−11	−10.4	17	11	8	2
23	−11	−10.7	15	11	2	2
Acarbose	−7.9	−7.7	5	3	5	3

**Table 3 molecules-27-03888-t003:** MD trajectory values obtained for CD-59 and acarbose complexed with α-glucosidase.

MD Trajectory Values	Apoprotein	Protein–Acarbose Complex	Protein–ZINC02789441 Complex
RMSD	0.30–0.35 nm	0.35–0.40 nm	0.30–0.35 nm
Rg	2.35–2.40 nm	2.40–2.45 nm	2.35–2.40 nm
SASA	220–230 nm^2^	230–235 nm^2^	220–230 nm^2^
Ligand H-bonds	-	9	7

**Table 4 molecules-27-03888-t004:** MD trajectory values obtained for CD-59 and acarbose complexed with α-amylase.

MD Trajectory Values	Apoprotein	Protein–Acarbose Complex	Protein–CD-59 Complex
RMSD	0.25–0.30 nm	0.35–0.40 nm	0.25–0.30 nm
Rg	2.25–2.30 nm	2.30 nm	2.25–2.30 nm
SASA	185–190 nm^2^	185–190 nm^2^	185–190 nm^2^
Ligand H-bonds	-	7	6

**Table 5 molecules-27-03888-t005:** Virtual screening of selected coumarin derivatives from pharmacophore studies for α-glucosidase.

Coumarin Derivative	Name	Binding Affinity (kcal/mol)	Total No. of Intermolecular Interactions	Total No. of Hydrogen Bonds
32	ZINC13496808	−11.8	14	6
9	ZINC59502854	−11.6	14	5
52	ZINC15109170	−11.6	13	3
5	ZINC02789441	−11.4	15	4
20	ZINC12779729	−11.4	12	2
68	ZINC41013463	−11.4	13	4
16	ZINC01641963	−11	12	4
91	ZINC09781623	−11	13	5
108	Acarbose	−7.9	5	5

**Table 6 molecules-27-03888-t006:** Virtual screening of selected coumarin derivatives from pharmacophore studies for α-amylase.

Coumarin Derivative	Name	Binding Affinity (kcal/mol)	Total No. of Intermolecular Interactions	Total No. of Hydrogen Bonds
51	ZINC68601297	−11.4	4	-
7	ZINC03144061_1	−11.4	4	-
23	ZINC08721887_4	−11.4	4	-
24	ZINC08721887_5	−11.4	4	-
43	ZINC15799331_4	−11.4	4	-
44	ZINC15799331_5	−11.4	4	-
10	ZINC03144061_4	−11.3	4	-
11	ZINC03144061_5_1	−11.3	4	-
5	ZINC02929288	−10.6	8	1
28	ZINC08901231	−10.5	8	-
49	ZINC21884078_3	−10.1	14	-
55	ZINC09781623	−10.1	13	3
9	ZINC03144061_3	−10	13	-
42	ZINC15799331_3	−10	14	-
46	ZINC20143660	−10	9	1
50	ZINC25576471	−10	18	4
8	ZINC03144061_2	−9.7	14	2
17	ZINC05350534_1_2	−9.7	14	2
21	ZINC08721887_2	−9.7	14	2
22	ZINC08721887_3	−9.7	15	-
31	ZINC13120435_2	−9.7	15	2
41	ZINC15799331_2	−9.7	14	2
48	ZINC21884078_2	−9.7	14	2
6	ZINC02933910	−9.6	13	6
18	ZINC05350534_2	−9.6	14	3
32	ZINC13120435_3	−9.6	15	-
52	ZINC02789441	−9.5	16	1
56	Acarbose	−7.7	2	2

**Table 7 molecules-27-03888-t007:** Selected potential dual inhibitors for α-glucosidase and α-amylase from the virtual screening of pharmacophore compounds.

Compounds	α-Glucosidase			α-Amylase		
	Binding Affinity (kcal/mol)	Total No. of Intermolecular Interactions	Total No. of Hydrogen Bonds	Binding Affinity (kcal/mol)	Total No. of Intermolecular Interactions	Total No. of Hydrogen Bonds
ZINC02789441	−11.4	15	4	−9.5	16	1
ZINC40949448	−10.5	14	5	−8.2	19	1
ZINC13496808	−11.8	14	6	−9.2	18	2
ZINC09781623	−11	13	5	−10.1	13	3
Acarbose	−7.9	5	5	−7.7	2	2

**Table 8 molecules-27-03888-t008:** MD trajectory values obtained for ZINC02789441 and acarbose complexed with α-glucosidase.

MD Trajectory Values	Apoprotein	Protein–Acarbose Complex	Protein–ZINC02789441 Complex
RMSD	0.40 nm	0.30–0.35 nm	0.30–0.35 nm
Rg	2.4 nm	2.4 nm	2.4 nm
SASA	220–235 nm^2^	220–235 nm^2^	220–235 nm^2^
Ligand H-bonds	-	7	9

**Table 9 molecules-27-03888-t009:** MD trajectory values obtained for ZINC02789441and acarbose complexed with α-amylase.

MD Trajectory Values	Apoprotein	Protein–Acarbose Complex	Protein–ZINC02789441 Complex
RMSD	0.30 nm	0.30 nm	0.25–0.30 nm
Rg	2.25–2.30 nm	0.30 nm	2.25–2.30 nm
SASA	185–190 nm^2^	185–190 nm^2^	185–190 nm^2^
Ligand H-bonds	-	6	7

**Table 10 molecules-27-03888-t010:** Results of binding free energy calculations obtained using MMPBSA technique.

Types of Binding Free Energy	ZINC02789441-α-Glucosidase Complex	Acarbose-α-Glucosidase Complex	ZINC02789441-α-Amylase Complex	Acarbose-α-Amylase Complex
	Values (kj/mol)	Values (kj/mol)	Values (kj/mol)	Values (kj/mol)
Van der Waal’s energy	−316.391 ± 15.473	−218.605 ± 23.706	−169.669 ± 17.479	−151.112 ± 19.233
Electrostatic energy	−21.871± 5.801	−4.761 ± 6.221	−6.992 ± 11.374	−10.911 ± 6.801
Polar solvation energy	106.897 ± 13.989	−103.307 ± 55.952	79.945 ± 50.793	61.951 ± 25.681
SASA energy	−22.576 ± 0.997	−17.835 ± 13.498	−12.899 ± 7.329	−14.929 ± 6.997
Binding energy	−115.796 ± 61.774	−137.894 ± 70.951	−213.410 ± 59.230	−112.119 ± 58.114

## Data Availability

Not applicable.

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
