# Peer review of "Discovery of Novel Coumarin Derivatives as Potential Dual Inhibitors against α-Glucosidase and α-Amylase for the Management of Post-Prandial Hyperglycemia via Molecular Modelling Approaches"

_molecules, 2022, doi:10.3390/molecules27123888_

Round 1

Reviewer 1 Report

The authors performed thorough study to explore potential inhibitors for α-glucosidase and α-amylase. The idea of having dual inhibitor for both enzyme is interesting and very useful with regard to clinical application.

The authors designed their experiment very well and they used the proper controls in each case.

In case of CD-59, the presence of one bromo substituent present at ortho position of phenyl ring of benzophenone, and methyl group is present at para position of benzoyl ring of benzophenone is characteristic feature of CD-59. I think it will be very useful and give more in depth understanding if the authors investigate the role of those two moities on the binding of CD-5 to α-glucosidase and α-amylase. This important for the future synthesis.

Reviewer 2 Report

I am a firm believer that in silico screening/ docking is the future for drug development. However this believe does not prevent me from expecting that any conclusion obtained in an in silico study must be validated with "real" experimental testing -like ITC, SPR, etc. This article is an example of what we should not do to credit in silico studies - that is the authors present a list of results from in silico without a single experimental result to confirm that the compounds do bind - in fact according to the authors they all bind in silico ..

In my opinion, without experimental data to add this work it is just a computational exercise - moreover with extensive lists and tables that make a very hard read.

Another point - it is not a strong argument to use as an example for a coumarin derived drug -  warfarin - that is used in the clinic ( a dangerous drug)  but is used also as a rat poison...
